# Oceanic upper crustal accretion by melt sill and lava flow interaction at Axial volcano

Han Wu [1,2], Wenxin Xie[1], Satish C. Singh [1] ✉, Hélène Carton [1], Graham M. Kent[1,3], Adrien F. Arnulf [4] & Alistair J. Harding[5]

Magmatically-accreted upper crust along mid-ocean ridges is traditionally considered to consist of lava flows overlying a sheeted-dyke complex. However, how the upper crust is formed at hotspot-influenced ridge segments remains unknown. Using three-dimensional seismic reflection data, here we report images of lava flow layering beneath Axial volcano on the Juan de Fuca Ridge. These lava flow layers dip inward towards the caldera and rift zones, resulting from subsidence caused by magma withdrawal beneath the summit and rift zone extension. Furthermore, our images reveal that the classic, laterally continuous sheeted-dyke complex is absent. Instead, lava flow layers are rotated and brought into direct contact with the magma domain, leading to partial assimilation of these units. Three-dimensional images also show that melt is injected outward along lava flow layers, suggesting that the upper crust at Axial volcano is formed by complex interactions between injected melt sills and lava flow layers.

The oceanic crust, which comprises two-thirds of the Earth's surface, is formed through partial melting of the mantle, where upwelling melt migrates towards the surface and accumulates in crustal magma reservoirs[1–3]. Some of this melt is erupted through feeder dykes, with erupted lava flows at the surface and intruded dykes forming the upper oceanic crust (Layer 2), while the residual melt cools through hydrothermal circulation and crystallises, forming the lower gabbroic crust (Layer 3)[4]. This architecture is the classic layer-cake oceanic crustal structure as described in the Penrose model[5]. Using ophiolite[6] and drilling studies as a framework[7,8], seismic investigations of the upper crust[9–12] have further divided it into Layer 2A and Layer 2B, where Layer 2A is interpreted to consist of lava flows and Layer 2B of sheeted dykes, although hydrothermal alteration has also been invoked as a possible mechanism explaining the location of the seismically-defined Layer 2A/2B boundary[13].

Although lava flows and sheeted dyke sequences can be unambiguously distinguished within ophiolites, drill cores, and tectonic windows[14,15], their in-situ characterisations using seismic methods have been difficult. This difficulty is because lava flow stratigraphy has not been seismically imaged thus far, and vertical or steeply dipping dykes are inherently difficult to image using either reflected or refracted waves. Furthermore, the boundary between lava flow and dyke sequences is not sharp, instead there is a high velocity gradient zone separating low-velocity lava flow layers above with the high-velocity dyke sequence below[4]. Therefore, the main methods to locate the Layer 2A/2B boundary has been the stacking of wide-angle energy in multi-channel seismic (MCS) data produced by the high velocity gradient at Layer 2A/B boundary[11] (Fig. 1a) or its identification is based on velocity contours and velocity gradients obtained using tomographic methods[16] (Fig. 1b). In order to image lava flow stratigraphy, three-dimensional (3D) pre-stack depth migration (PSDM) methodology is required to adequately refocus the seismic energy in the shallow crust (e.g., Fig. 1c), which we have carried out using 3D MCS data from Axial volcano[17].

Axial volcano lies at the intersection of the intermediate-spreading Juan de Fuca Ridge and Cobb hotspot (Fig. 2) and seems to have been formed in the last ~0.5 Ma[18–20]. It hosts several hydrothermal fields[21,22] and has been the site of three documented eruptions

[1]Université Paris Cité, Institut de Physique du Globe de Paris, CNRS, Paris, France. [2]School of Earth Sciences and Engineering, Sun Yat-sen University, Zhuhai, China. [3]Nevada Seismological Laboratory, MS-0174, University of Nevada, Reno, NV, USA. [4]Amazon, San Diego, CA, USA. [5]Cecil H. and Ida M. Green Institute of Geophysics and Planetary Physics, Scripps Institution of Oceanography, University of California San Diego, La Jolla, CA, USA. ✉e-mail: singh@ipgp.fr

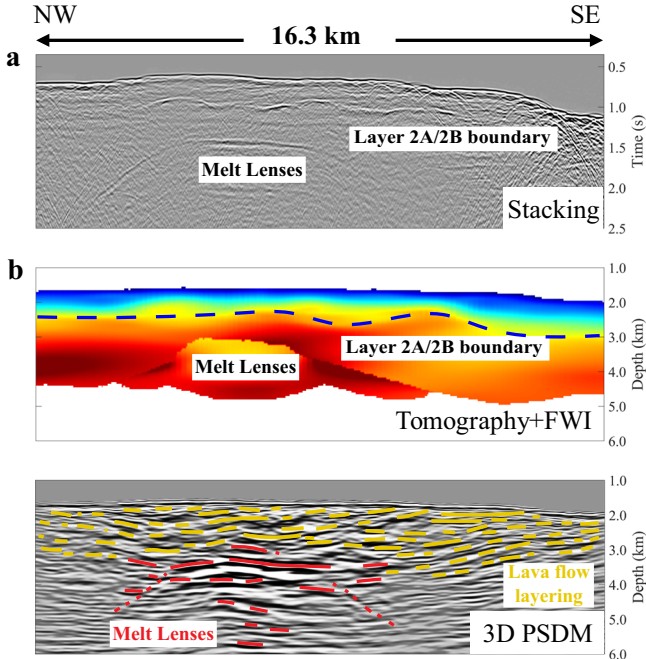

**Fig. 1 | Different representation of seismic images. a** Seismic image obtained using the conventional stacking method showing Layer 2A and melt lens reflections (e.g., Harding et al.[11]) along a 2D profile within the study area. The location of the profile is shown in Supplementary Fig. 1. **b** The P-wave velocity obtained using a combination of travel time tomography and full waveform inversion (FWI) of same data set[16]. The blue curve marks a change in the velocity gradient, interpreted as the base of Layer 2A. **c** 3D pre-stack depth migrated (PSDM) image along a coincident profile (Xline 2356) showing lava flow layering (light yellow) and melt lenses (red) (see below). Note the absence of a clear Layer 2A/2B boundary in the PSDM image.

(1998, 2011, 2015)[23–26] with the next one considered imminent (as of late 2025) on the basis of geodetic-derived deformation and seismicity rates[27]. Axial volcano lies at a water depth of about 1.4 km; it displays a nearly flat, 20–40 km wide summit hosting a horseshoe-shaped 8 km × 3 km caldera that separates two rift zones: the north rift zone (NRZ) and south rift zone (SRZ) (Fig. 2)[18,28]. The current caldera is interpreted to have been formed 1259 +/− 119 years BP, based on the ages of large volume lava flows on the distal SRZ[29]. The volcano is continuously monitored using a sea-to-land, fiber-optic cabled observatory, recording earthquake, geodetic, heat flow and chemical data[30].

In 1998, sparse wide-angle ocean-bottom seismic refraction data were acquired that showed the presence of a large magma body embedded within an up to 11-km-thick oceanic crust[31]. A 2D MCS dataset collected shortly thereafter in 2002 revealed complex magma sill reflections and a Layer 2A/2B boundary[16,32,33], similar to images obtained along non-hotspot-influenced mid-ocean ridges (e.g., Fig. 1a, b)[9–12]. Based on an updated tomographic velocity model that combined prior on-bottom wide-angle[31] and redatumed, 6-km-long 2D streamer data[16], the Layer 2 A thickness was found to vary between 0.4 km and 1.4 km, with the thinnest Layer 2 A (0.4−0.7 km) observed just outside of the caldera with a slightly thicker Layer 2A (0.75−0.95 km) centered beneath the caldera[32]. However, as tomographic images only provide low-resolution information, it is not possible to decipher the fine-scale lava flow layering of this uppermost oceanic crust and investigate relationships to the underlying magmatic system.

In 2019, an exceptional 3D MCS dataset was acquired at Axial volcano (Fig. 2; Supplementary Fig. 1), showing the presence of four funnel-shaped melt boundaries beneath the volcano[17], interpreted as the top of a magma domain[17], within which melt sills are distributed in a matrix of possibly crystal mush, and representing the crustal lithosphere-asthenosphere boundary (LAB).

Here, we processed the same MCS dataset differently using 3D Kirchhoff PSDM (see Methods), instead of post-stack time migration[17]. Our processed 3D seismic volume provides the clear images of lava flow stratigraphy within the upper crust and its relationship to the complex underlying magma domain, suggesting that the upper oceanic crust is formed by a complex interaction between melt sills and the extrusive layer within a ridge-hotspot environment.

## Results and discussion

### Lava flow layering

The 3D PSDM reveals lava flow layering between the seafloor and the magma domain (Supplementary Movies 1 and 2). A block diagram (Fig. 3, Supplementary Fig. 2 and Movie 3) shows lava flow layering beneath the northern (Inline 341) and southern (Inline 178) rift zones, and along a perpendicular profile (Xline 2601) through the caldera. We identify the lava flow layering based on its reflection characteristics and seismic attributes (Supplementary Fig. 3, see Methods). The top of the highly reflective zone, interpreted as a volume containing melt sills, defines the top of the magma domain, representing the lithosphere-asthenosphere boundary (LAB) (Figs. 2–4)[17]. Although the term LAB is generally used elsewhere to define the base of the lithosphere in the upper mantle, here a crustal melt sill beneath the ridge axis separates solid brittle crust above with partially molten lower crust and upper mantle below, and therefore the top of the magma domain at Axial volcano is termed as the LAB[17].

North of the caldera, lava flow layers from the southeast dip toward the NRZ before becoming flat beneath the NRZ (Fig. 3). Similarly, south of the caldera, lava flow layers dip gently inwards toward the rift zone, indicating a symmetric pattern. The perpendicular Xline profile shows that the layers continue beneath the caldera, but in some cases are offset by small-throw steeply dipping faults with vertical offsets of tens of metres. Four Xlines spaced 5 km apart across the caldera exhibit a similar pattern of lava flow layering in the upper crust, with a pervasive inward-dipping geometry towards the magma domain beneath the caldera (Fig. 4, Supplementary Fig. 4 and Movie 4). As the dip of the layers increases with depth (Fig. 5), some layers have deepened up to 1 km through rotation, with the overall stack of lava flow layers reaching >3 km in thickness. The pattern of inward-dipping lava flow layers towards the magma domain extends southeast of the caldera as seen along Xline profile 2424 (Fig. 4). The dip of these layers varies, depending upon the location within the 3D box, ranging from nearly flat just below the seafloor to dips of up to 18° near the magma domain (Fig. 5). Sometimes, these layers flatten as they approach the magma domain and beneath the rift zones. Rift zone-coincident profiles were extracted from the 3D PSDM volume (Fig. 6, Supplementary Fig. 5), which highlight the relationship between lava flow layering, faults, and the underlying magma domain. Although the images along the NRZ are relatively poor, layered structures are present. In contrast, layering is very prominent along the SRZ profile down to 3 km below the seafloor (bsf). The analysis of the 3D PSDM volume indicates that the upper crustal layered stratigraphy is present over more than 85% of the study area; only the northeast corner of the 3D box does not exhibit strong layering. Given the frequency bandwidth of our seismic images (5–30 Hz), individual lava flows cannot be imaged, but rather the visible layering likely represents packages of flows, with vertical resolution dependent on wavelength. The vertical resolution is generally a quarter of the wavelength, which for an upper crustal velocity of 3–5 km/s would be ~25–250 m (Supplementary Fig. 3). Wide-aperture (6 km source-receiver offset) 3D imaging allows us to image structure on similar lateral scales (250 m). Although the nature of the impedance contrast that produces these reflections is unknown, the changes in rock properties of an individual flow, the residence time at

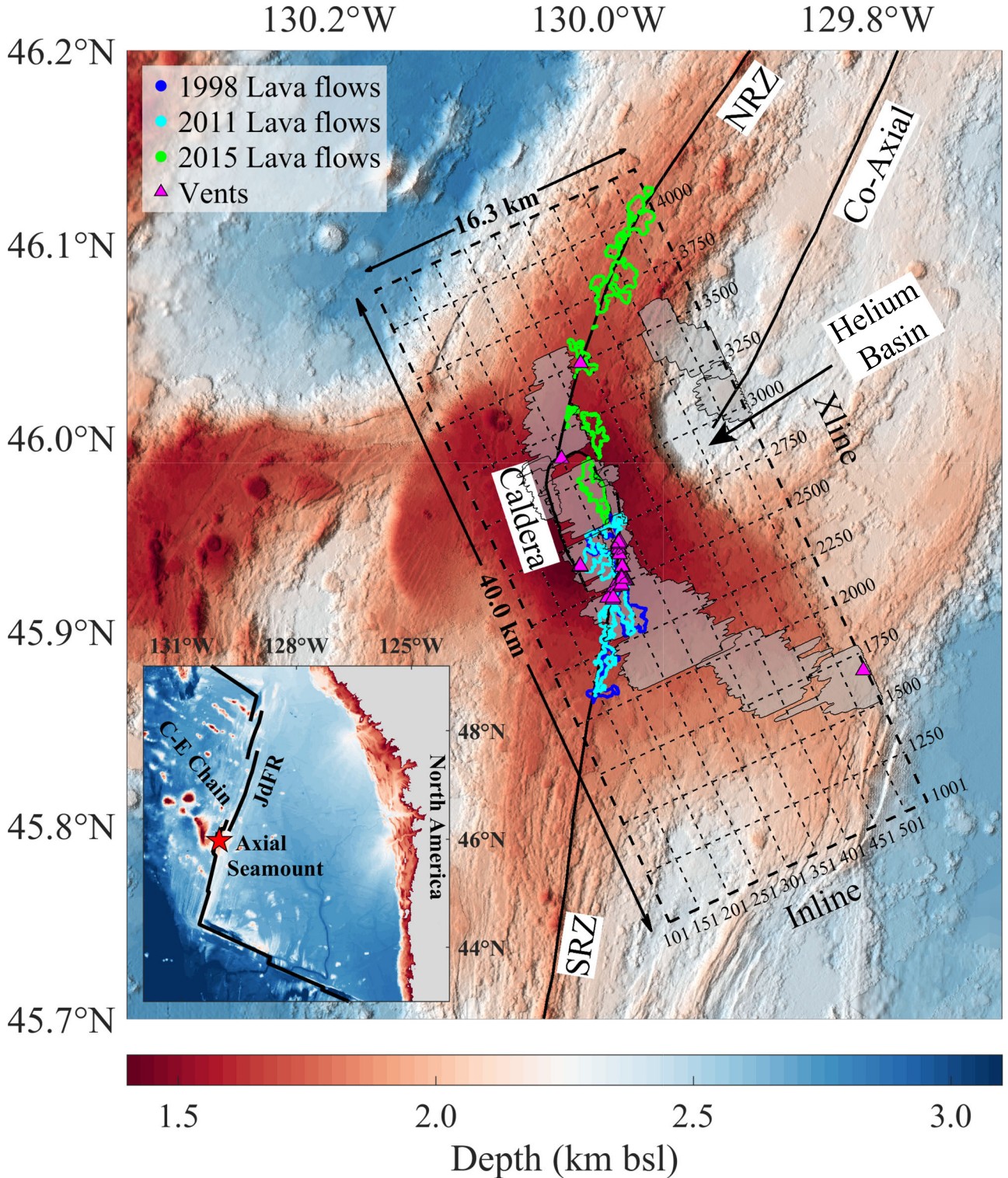

**Fig. 2 | Bathymetric map of Axial volcano.** The map shows the north rift zone (NRZ), south rift zone (SRZ), hydrothermal vents (magenta-filled triangles)[21,22] and the lava flows from the last three eruptions in 1998 (blue), 2011 (cyan) and 2015 (green)[23–26]. The horseshoe-shaped caldera, 8 km × 3 km, is also highlighted. The transparent grey-shaded regions indicate the funnel-shaped upper boundary of the magma domain, interpreted as the lithosphere-asthenosphere boundary (LAB), surfaces extracted from a 3D time migration image[17]. The thick dashed rectangle indicates the area of the 3D seismic reflection box and thin dashed lines mark the inline and crossline numbers. The Inline numbers range from 101 to 538 spaced at 37.5 m and the cross lines (Xlines) range from 1001 to 4200 spaced at 12.5 m, giving rise to a 16.3 km × 40 km 3D box. The origin of the 3D box is at the southwest corner. Inset map shows the Northeast Pacific region. The star shows the location of Axial volcano. bsl Below sea level.

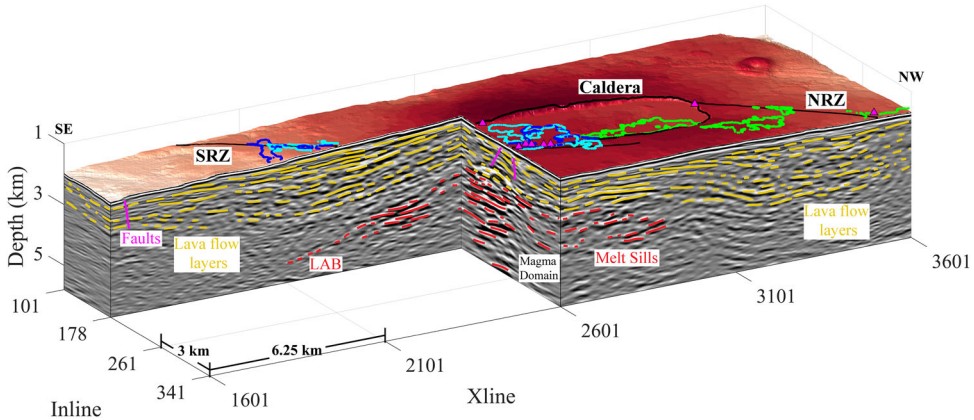

**Fig. 3 | Lava flow layers in the 3D seismic survey.** Block diagram showing Inline (178 and 341) and Xline (2601) seismic reflection images crossing the south rift zone (SRZ), north rift zone (NRZ), and the south end of the summit caldera. Interpreted lava flow layers are highlighted using yellow lines, magma domain sills (below the LAB) are marked as red lines, and faults are shown as magenta lines. Locations of hydrothermal fields are shown as small, magenta-filled triangles[21,22]. Outlines of the 1998 (blue), 2011 (cyan) and 2015 (green) eruption lava flows are overlain on the bathymetric map[23–26]. The locations of seismic profiles are shown in Supplementary Fig. 1; the non-interpreted seismic profiles are shown in Supplementary Fig. 2.

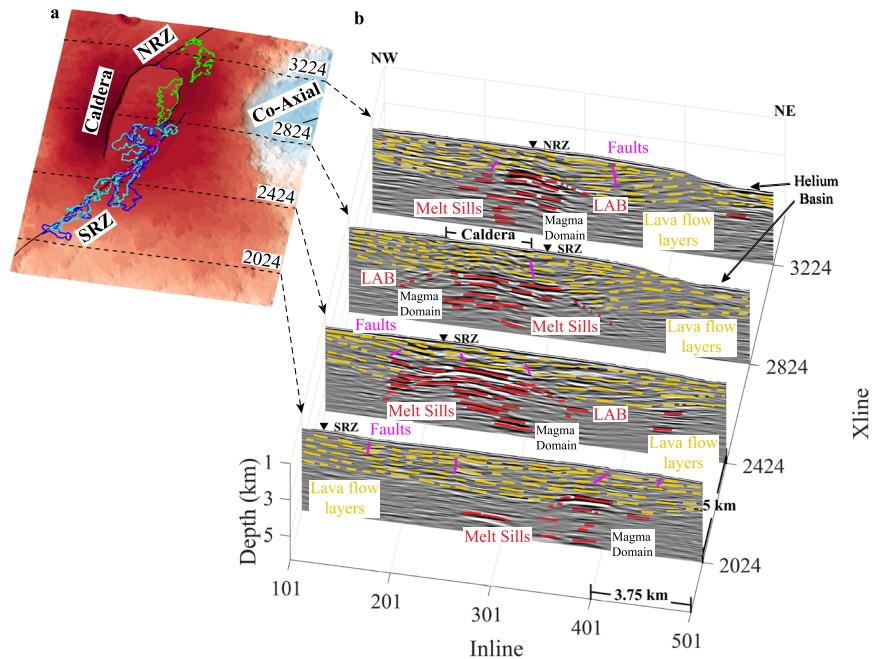

**Fig. 4 | Selected Xlines crossing sections. a** The locations of four representative Xlines (5 km apart). SRZ: South rift zone. NRZ: North rift zone. Locations of hydrothermal fields are shown as small, magenta-filled triangles[21,22]. Outlines of the 1998 (blue), 2011 (cyan) and 2015 (green) eruption lava flows are overlain on the bathymetric map[23–26]. **b** Four Xline seismic images are shown with interpreted lava flow stratigraphy (yellow), magma domain sills (red) and faults (magenta). LAB Lithosphere-asthenosphere boundary. The locations of the lines are shown on the map on the left. Xlines 2024 and 2424 are southeast of the caldera, Xline 2824 crosses the central caldera, and Xline 3224 is located northwest of the caldera. The non-interpreted seismic profiles are shown in Supplementary Fig. 4.

the seafloor where seawater alteration occurred, and/or porosity contrasts, likely yields this contrast.

## Magma domain and melt sill injection

Using images from the 3D post-stack time-migrated volume, Kent et al.[17] observed four large-scale funnel-shaped boundaries that define the top of a magma domain. The 3D PSDM imagery presented here confirms those features but also reveals other smaller melt sills of various sizes outside of the identified magma domain[17], which suggests that the influence of the magma domain may extend further east of the caldera (Fig. 7, Supplementary Fig. 6 and Movie 5). For example, within profile Inline 373 shown in the bottom left of Fig. 7, there are six sill-like melt bodies imaged; but only two of the most southeasterly melt sills

were previously reported[17]. A critical observation is that the lava flow layers not only abut the sides of the magma domain but, in some locations, melt appears to be injected outwards between lava flow layers above the magma domain (Fig. 8, Supplementary Fig. 7). However, it is sometimes difficult to distinguish between the lava flow layers, injected melt sills between lava flow layers, and small melt sills that form the top of the magma domain. Here, we distinguish intruded melt sills from lava flow layers based on their strong reflectivity, large instantaneous amplitude, instantaneous phase with positive to negative transition, lower frequency content, complex shapes, stratigraphic relationships, and depth (Supplementary Fig. 3). For example, along lava flow layers, if the reflection becomes brighter over a very short distance, is characterised by lower frequencies, and abuts the magma

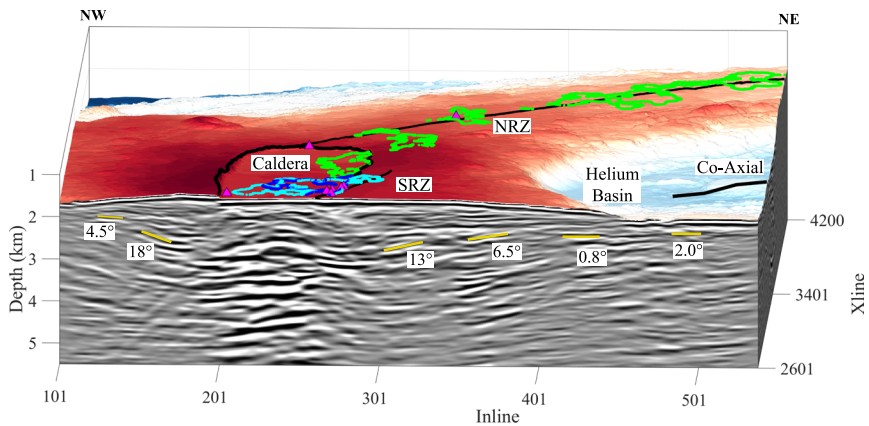

**Fig. 5 | Seismic image across Axial caldera.** A block diagram highlights Xline 2601 that crosses the southeast portion of the caldera. The dips of lava flow layers (yellow) are marked. The location of this seismic profile is shown in Supplementary Fig. 1.

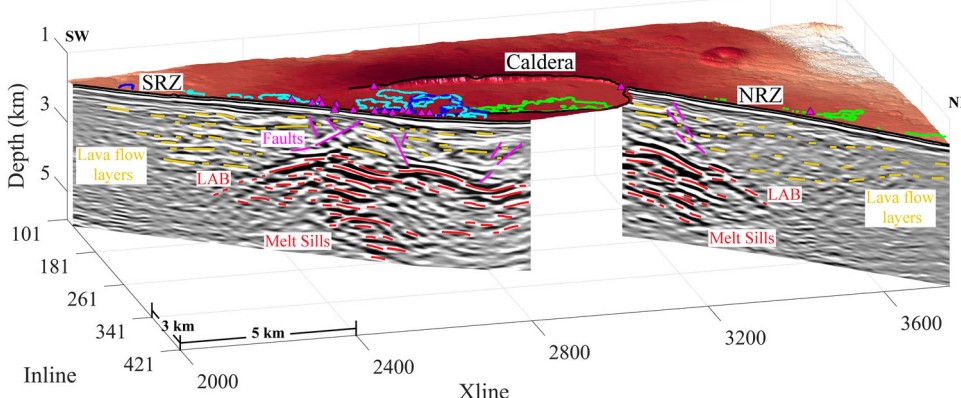

**Fig. 6 | Seismic images along the rift zones.** Seismic images along the rift zones (SRZ, NRZ) (light blue lines in Supplementary Fig. 1). SRZ South rift zone, NRZ North rift zone. Yellow lines are lava flow layers, red lines show magma domain sills, and magenta lines denote upper crustal faults. LAB: Lithosphere-asthenosphere boundary. Locations of hydrothermal fields are shown as small, magenta-filled triangles[21,22]. Outlines of the 1998 (blue), 2011 (cyan) and 2015 (green) eruption lava flows are overlain on the bathymetric map[23–26]. The non-interpreted seismic profiles are shown in Supplementary Fig. 5.

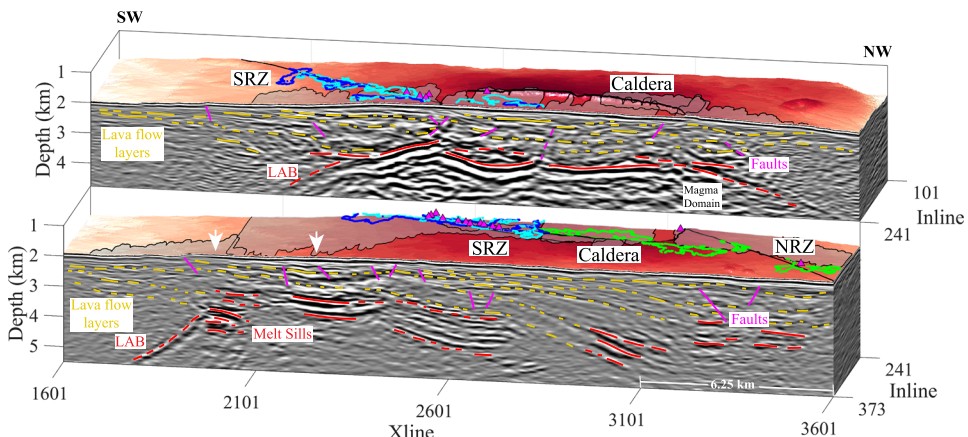

**Fig. 7 | Lava flow melt lens interaction.** Block diagram showing Inline (241 and 373) seismic reflection images. Inline profile 373 (front) is located northeast of the caldera and shows lava flow layers interacting with magma domain melt sills at six locations over the 25 km distance along the profile. Lava flow layers are highlighted in yellow, magma domain sills in red, and faults in magenta, white arrows indicate the two magma domain magma sills reported previously[17]. SRZ South rift zone, NRZ North rift zone, LAB Lithosphere-asthenosphere boundary. Inline 241 (back) shows lava stratigraphy above the magma domain beneath the caldera. Locations of hydrothermal fields are shown as small, magenta-filled triangles[21,22]. Outlines of the 1998 (blue), 2011 (cyan) and 2015 (green) eruption lava flows are overlain on the bathymetric map[23–26]. The non-interpreted seismic profiles are shown in Supplementary Fig. 6.

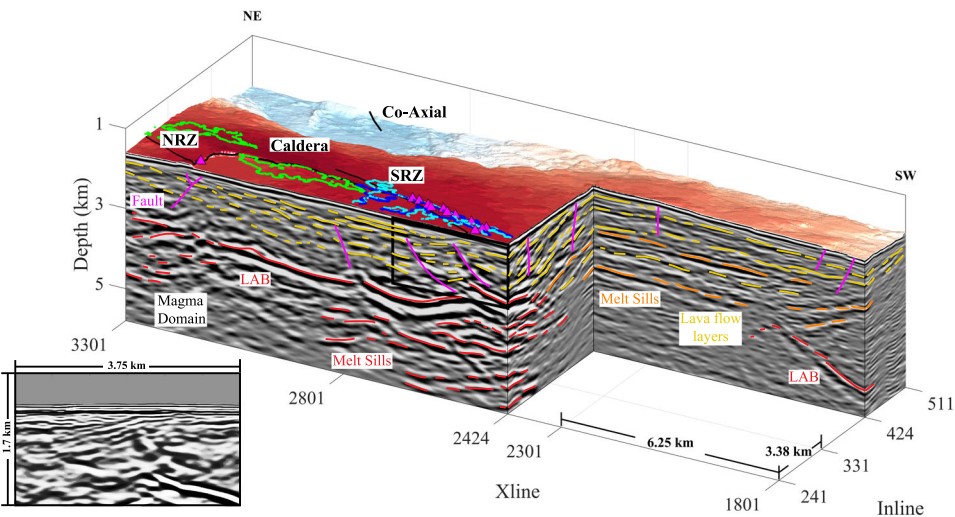

**Fig. 8 | Melt sills injected along lava flow layers.** Block diagram showing Inline (241, 424) and Xline (2424) seismic reflection images. Inline 241 highlights lava stratigraphy and curved faults above the magma domain beneath the caldera. The location of inset (bottom left corner) is shown by black box in upper right of Inline 241. Inline 424 and Xline 2424 highlight melt sills injected along lava flow layers. Lava flow layers are highlighted in yellow, magma domain sills in red, injected sills in orange, and faults in magenta. Injected melt sills have different orientations with some dipping southwest back toward the caldera (right edge of Xline 2424), while others dip to the northwest (right edge of Inline 424). SRZ South rift zone, NRZ South rift zone, LAB Lithosphere-asthenosphere boundary. Locations of hydrothermal fields are shown as small, magenta-filled triangles[21,22]. Outlines of the 1998 (blue), 2011 (cyan) and 2015 (green) eruption lava flows are overlain on the bathymetric map[23–26]. The non-interpreted seismic profiles are shown in Supplementary Fig. 7.

domain downdip, this event is interpreted as an upward injected melt sill (Fig. 7, Supplementary Fig. 6). Moreover, as these melt sills cool within the lava flow layers, the interface would still produce a strong impedance contrast that enables imaging a distinct reflection as opposed to the typical lava flow layers observed closer to the seafloor (Fig. 6, Supplementary Fig. 5).

Lava flow layering within the caldera above the magma domain is more chaotic than outside the caldera, apparently disrupted by magmatic and hydrothermal processes (dyking, sill injection) or by faulting and block rotation during caldera collapse events (Fig. 8, Inline 241). Lava flow layering, faults and rotated layers are clearly visible on seismic images (Fig. 7, Inline 241). Some of the shallow faults above the magma domain are distinctly imaged, with curved geometries and rotated layers (Fig. 8, inset). Away from the caldera, one can clearly see interaction between lava flow layers and melt sill intrusions (Fig. 7, Inline 373; Fig. 8, Inline 424). On the other hand, vertical dykes and hydrothermal conduits are inherently difficult to image.

## Subsidence
Mapping of lava flows from the past three eruptions[23–26] shows that these events initiated from dyke intrusion along the eastern edge of the caldera[34], possibly sourced from the shallowest part of the magma domain[17]. From that nucleation point, the dykes propagated either north or south along the rift zones for distances up to 20–30 km from the caldera with lava outpouring from fissures along the rifts[24–26]. Within the caldera, recent sheet flows have thicknesses of 3–5 m near eruptive features increasing to ~15 m down the slope[25,26,35,36]. Away from the caldera, lava flows erupted along the rift zones form pillow mounds and ridges with thickness reaching >120 m[24,26]. Even though the seafloor smoothly dips away from the caldera, the seismically imaged lava flow layers show a gentle dip either towards the magma domain, rift zones, and/or the caldera. The lava flow piles thicken as they approach the rift zones and the underlying magma domain, with the largest subsidence evident along the rift zones and within the caldera. This geometry suggests that the lava flow layers have subsided after their formation. There are two likely mechanisms that can cause the observed localised subsidence: (1) extension along rift zones, and (2)

emptying of magma reservoirs during occasional large-volume eruptions that cause caldera collapse.

As the lava flow layering in the seismic imagery generally thickens toward the caldera and rift zones, it is likely that this pattern is simply the result of infilling of space caused by the subsidence. However, the terminations of the deepest imaged lava flow layer are not always located beneath the rift zones; instead, some are shifted either towards the caldera or the large melt funnels[17]. The horizontal wavelength of subsidence is on the scale of 10–15 km, and such long-wavelength subsidence is likely due to the cumulative effects of major withdrawal events from the sub-caldera magma domain (Fig. 9).

Ridge-ward dipping lava flow layers have also been observed in Iceland, both east and west of the ridge axis[37,38]. There too, the dip of lava flow layers increases with depth from nearly horizontal at shallow depth to 5–10° over 10–15 km distance. Drilling results indicate that lava flow layers that lie at 1.5–2.0 km depth were deposited on land or in shallow water environment[39,40]. Gradual sagging of lava flow layers due to continuous loading of lava flows in the rift zone was invoked to explain deepening of lava flows towards the rift zone for Iceland[37,41], but here we show that emptying of magma reservoir during large volcanic eruption might be the main process for the increasing dips towards the rift zones and caldera. Seaward dipping reflections (SDRs) commonly observed along volcanic continental margins[42,43] resemble the dipping lava flow reflections observed at Axial volcano. Taken together, the lava flow layering in Iceland and at Axial volcano may represent a special class of crustal accretion at ocean spreading centres occurring along magmatically enhanced sections of mid-ocean ridges, such as those influenced by ridge-plume interaction[41,44,45].

## Assimilation
On the basis of seismic reflection images of the funnel-shaped upper surface of the magma domain observed down to 6 km depth, Kent et al.[17] hypothesised that assimilation of dykes and gabbroic rocks has been ongoing, as a consequence of a phase of enhanced magma supply currently taking place at Axial volcano. In addition to evidence of melt sill injection along lava flow layers, our images show that some of these inward-dipping lava flow layers are in direct contact with the magma

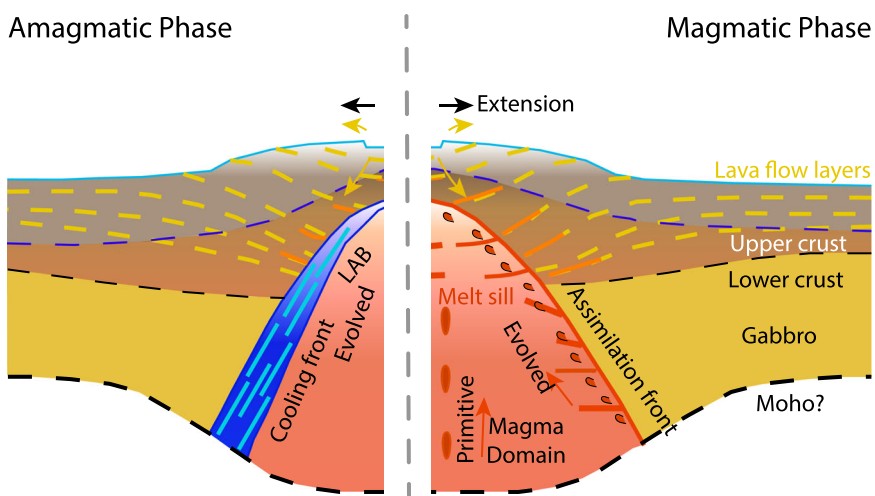

**Fig. 9 | A model of crustal accretion at plume-influenced ridges.** Schematic diagram showing the crustal accretion by injected melt sill and lava flow interaction during a strong magmatic phase (right). Yellow curves indicate subsiding lava flow layers injected by melt sills (in orange). Assimilation occurs at the lithosphere-asthenosphere boundary (LAB, in red). Blue dashed line indicates the possible domain boundary (Figs. 4, 7 and 8). location of the classical Layer 2A/2B boundary during the magmatic phase. Note the absence of a sheeted dyke sequence. In contrast, when magmatism wanes (left), the LAB becomes an accreting boundary as magmas become less evolved. Blue and light blue lines indicate gabbroic accretion.

If these lava flow layers were initially emplaced at the seafloor, they would have had higher porosities[46,47] filled with water[39,40]. As they get buried, heated, and altered[40], they eventually lie in contact with the top of the magma domain. The melting temperature of hydrated lava flows is ~800 °C, much less than that of primitive basalt (>1100 °C)[48], therefore the contact zone between hydrated lava flow layers and melt within the magma domain could constitute a zone of interaction with assimilation of the lava flow layers directly into the magma domain[48] (Fig. 9). Although some primitive lavas erupted during the 2015 eruption, most of the lava emplaced during the three historical eruptions is somewhat chemically evolved[36]. At this time, there is no direct geochemical evidence of assimilation in the basalt chemistry of erupted lava flows at Axial volcano, the presence of a young andesite ridge along the NRZ north of Axial volcano[49] suggests potential remelting and assimilation and/or a high degree of fractionation[49], possibly supporting the assimilation hypothesis.

Elsewhere, the assimilation of some of the sheeted dyke sequence into the underlying magma chamber has been documented from ophiolites and laboratory experiments[48,50], but the possible assimilation of basaltic lava flows has not been considered until now. Iceland is an archetypical example for the observation of remelting and assimilation of crustal rocks, with at least 10% of lavas of rhyolitic origin[51,52]. Rhyolitic melt was unexpectedly encountered at 2.1 km depth during the Krafla Testbed Drilling project in 2009[53,54]. Although several hypotheses have been proposed for the origin of these rhyolites, melting and assimilation of wall rock[51] or partial melting of hydrated and hydrothermally altered crust[53] remain the main hypotheses. Assimilation has also been reported in the Oman ophiolite[4,48], at a slow spreading ridge[55] and at an ultra-slow spreading ridge[56] and hence assimilation might be more widespread along the mid-ocean ridge system than previously realised, especially at sites with enhanced magma supply, such as those located at a ridge-plume intersections.

## Hydrated-dehydrated boundary

So where is the sheeted-dyke complex? Although some vertical dykes must be present above the magma domain and along the rift zones, the imaging of thick lava flow layers, some of which are in direct contact with the magma domain, suggests that the classic thick sheeted dyke sequence of Layer 2B is absent beneath Axial volcano, and the crust predominantly consists of lava flows and gabbro. This observation is at odds with previous studies of the upper crust at Axial volcano[16,32], where the upper crust was inferred to consist of two layers: lava flows and sheeted dyke complex. Instead, the 3D seismic imagery presented here suggests that the upper crust is mostly formed by lava flows and melt-sill interaction. It is certain that dykes feed magma to the surface during eruptions, but a combination of dyke assimilation and lateral sill intrusions within the lava flow layers may erase the evidence of the dyking process. However, since dykes are the main conduits of erupted lavas, extending for tens of kilometres along the rift zones, a dyke sequence might be present along the rift zones away from the summit of Axial volcano and along the non-hotspot-influenced ridge segments of the Juan du Fuca Ridge.

Based on a comprehensive tomographic study[16], it was suggested that the maximum thickness of Layer 2A (i.e., presumed lava flows) was ~1.4 km. However, 3D seismic reflection images show lava flow layering down to >3 km depth bsf, suggesting that the conventional Layer 2A/2B boundary does not represent a lava flow–sheeted dyke boundary beneath Axial volcano. Instead, it may correspond to the depth of melt sill injection into the lava flows. We envision that when hot molten rock (melt sills) is injected into the upper crustal lava flow layers, the heat would metamorphose and dehydrate the nearby lava flow layers. This would make the rock denser and increase the contrast in seismic velocities. Over time, as more melt sills are injected and solidify, the seismic velocity differences between the altered lava flow layers and the solidified melt sills would gradually diminish. The combined effect would progressively increase the seismic velocities of deeper layers, which could explain the velocity boundary that creates the classical Layer 2A/2B seismic reflections observed at Axial volcano (Figs. 1b and 10).

Such intrusion of melt sills into lava flow layers has also been observed in the lower sequences of Icelandic lava flows[57], and in SDRs along volcanic continental margins[58]. Subsequent metamorphism (in the greenschist and amphibolite facies) leads to a significant increase in the P-wave velocity at this contact surface[58]. To further investigate the transition between shallow lava flows and their deeper intruded counterparts, seismic full waveform inversion (FWI) was performed along one of the long-offset 2D profiles (having a maximum offset of 12 km, see Methods), to determine the high-resolution P-wave velocity structure of the upper crust (Fig. 10). By combining conventional imaging and FWI results, we find that the conventional base of Layer 2A from stacking of wide-angle reflection data corresponds to a high

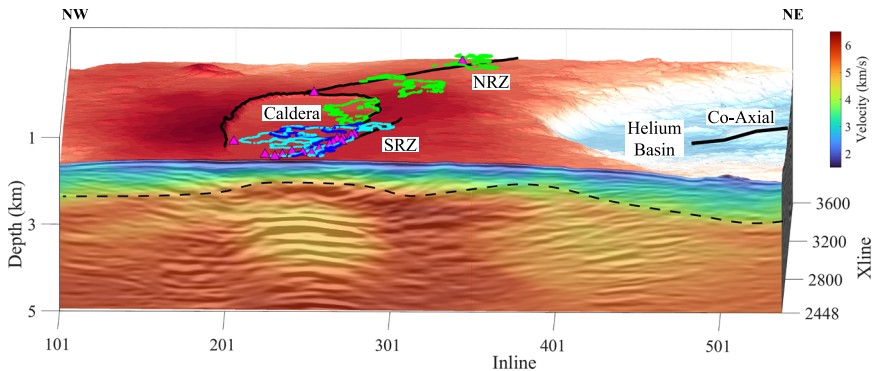

**Fig. 10 | Combined full waveform inversion (FWI) result and 3D PSDM image.** The FWI was performed along 2D profile (2003), which is coincident with Xline 2448. The FWI results are in colour and seismic PSDM image in black and white. The conventional Layer 2A/2B boundary is indicated by black dashed line and corresponds to high velocity gradient in the FWI results. The locations of hydrothermal fields are shown as small, magenta-filled triangles[21,22]. Outlines of the 1998 (blue) and 2015 (green) eruption lava flows is overlain on the bathymetric map[23–26].

velocity gradient zone, where the P-wave velocity increases from 4.1 km/s to 6.0 km/s at a depth of ~1 km below the seafloor, with a decrease in porosity from 18% to 4% (Supplementary Fig. 8)[46,47]. This rapid vertical change in P-wave velocity within the lava flow stratigraphy supports the hypothesis that the Layer 2A/2B boundary beneath Axial volcano may be a dehydration/metamorphic boundary.

Our findings support an interpretation that the shallow crust at Axial volcano differs from the classic model of magmatic crustal accretion at mid-ocean ridges. We interpret that the upper crust at Axial volcano does not consist of the expected lava flows overlying a sheeted dyke complex, but instead, its formation is primarily governed by the interaction between lava flows and melt sills injected into the lava flow layers and by subsequent metamorphism. Thus, these insights into the active shallow processes at Axial volcano provide a template for understanding the evolution of ridge segments or rift zones dominated by a central volcano, including hotspot-influenced ones, as well as large igneous provinces such as Iceland.

Our shallow crustal observations complement the deep crustal LAB imaging of Kent et al.[17] from the same dataset. Together, these studies suggest that Axial volcano represents an endmember of crustal accretion during strong magmatic phases, where assimilation along a thermally-controlled LAB[17] and melt sill injection into lava flow layers, reported in this study, dominate over classical dyke-dominated accretion (Fig. 9). As magmatism wanes, this system may revert to a more classical crustal architecture.

## Methods
### Data acquisition
In 2019, 3D multi-channel seismic (MCS) data were acquired during the Axial-3D Expedition (NSF-funded cruise MGL1905) aboard the R/V Marcus Langseth. The acquisition was designed to provide high-resolution subsurface imaging of the Axial volcano region, using a state-of-the-art seismic array with careful planning to ensure high data quality and complete 3D coverage.

**Acquisition geometry.** The seismic acquisition system consisted of four 5.85-km-long hydrophone streamers spaced at 150 m. These streamers were equipped with a total of 468 channels per streamer, spaced at 12.5 m intervals. The configuration allowed for dense spatial sampling, minimising spatial aliasing and providing comprehensive data coverage across the survey area (Supplementary Fig. 9).

The source array featured two flip-flopping airgun arrays, each with a total volume of 54.1 L. These arrays were towed at a spacing of 75 m across the ship's centreline, providing an alternating source configuration that increased acquisition efficiency (Supplementary Fig. 9).

The depths of the source and receivers were carefully chosen to optimise signal fidelity and minimise noise. The source depth of 12 m was selected to enhance the low-frequency energy output of the airguns, crucial for imaging deeper geological structures. The streamer depth of 16 m provided a balance between signal strength at lower frequencies and suppression of surface-generated noise.

**Survey area and coverage.** The survey covered a 40 km by 16.3 km area (Fig. 2, Supplementary Fig. 1) constructed from 53 primary sail-line passes to ensure high-quality baseline coverage. To minimise data gaps caused by navigation or acquisition challenges, 10 additional Inline passes were performed, strategically placed to improve coverage uniformity. Furthermore, 6 lines were reacquired to address areas with incomplete data, ensuring comprehensive and reliable coverage throughout the entire study area. These data were binned at 12.5 m × 37.5 m, providing 3200 Xlines (1001–4200) and 438 Inlines (101–538), within a framework where the Inlines were spaced at 37.5 m and Xlines at 12.5 m (Supplementary Fig. 9). The fold (numbers of traces in each bin) was around 78 (Supplementary Fig. 9d), which provided a good signal to noise ratio after processing.

**Data quality measures.** To ensure high-quality data acquisition, real-time quality control was conducted during the cruise. This included monitoring the positions of streamers and sources, analysing navigation accuracy, and continuously evaluating the recorded data for noise or signal issues. The multi-pass design and reshoot strategy further ensured data continuity and completeness, reducing uncertainties in the final seismic dataset.

### Initial velocity model construction
An initial interval velocity model, derived from a prior high-resolution 3D tomographic inversion study[16], was used for post-stack time migration imaging, and was the subject of a recent publication by Kent et al.[17]. This model accurately captured the large-scale subsurface velocity structure, providing a reliable baseline for further refinement. The model, centred on the southeast edge of the Axial volcano caldera (45.92°N, 129.99°W), covered a 30 km × 40 km area, rotated 12.9°N, with a grid interval of 100 m (Supplementary Fig. 10a). This velocity model was downloaded from the Marine Geoscience Data System (MGDS)[16]. However, this velocity model contained regions where velocity values were set to NaN (Not a Number) due to insufficient ray coverage. To adapt this model for our study, a series of steps was undertaken to rectify this problem, improve the resolution, and adapt its applicability within the study area (Supplementary Fig. 10).

**Gap filling.** A specialised interpolation algorithm[59] was applied to replace missing data (NaN values) by extrapolating from neighbouring, non-missing data points. This ensured spatial continuity while maintaining the geophysical plausibility of the velocity field (Supplementary Figs. 10a, b).

**Out-of-bounds velocity assignment.** For regions within our study area, but outside the original velocity model's spatial coverage, an average one-dimensional (1-D) velocity profile beneath the seafloor was used to populate the velocity field. This approach provided reasonable velocities in the unmodelled areas (Supplementary Fig. 10b).

**3D smoothing.** 3D smoothing was employed to address abrupt velocity anomalies and reduce noise in the velocity model. This process smoothed irregular interfaces, while preserving key geological features, such as the melt lens and the Layer 2A/2B boundary (Supplementary Fig. 10b).

**Water layer.** The seafloor depth was constrained using high-resolution bathymetric data (https:// www.gebco.net). The velocity of the water column above the seafloor was uniformly set to 1.5 km/s.

**High-resolution resampling.** The velocity model was refined to match the resolution of the data acquisition geometry. Using 3D interpolation, the model was reconstructed with finer grid intervals of 12.5 m (Xline), 37.5 m (Inline), and 5 m (depth) (Supplementary Fig. 10c). This higher resolution velocity model ensured its compatibility with the seismic acquisition parameters and subsequently helped in improving the accuracy of seismic imaging.

## Data pre-processing

A systematic pre-processing workflow was employed to enhance signal quality and mitigate noise, addressing challenges specific to marine seismic datasets. Key steps are outlined below:

**Data interpolation.** Interpolation was performed to compensate for gaps caused by dead channels with zero seismic energy. A dip-coherency-based algorithm was utilised, ensuring that both flat and dipping events in shot gathers could be adequately interpolated. Supplementary Fig. 11 shows a gather before and after the data interpolation, showing that the dead traces have been replaced by live traces, with coherent energy.

**Swell noise attenuation.** The raw recorded data have low-frequency swell noise, which was removed using an expectation-maximisation algorithm[60], which statistically separates coherent signal from random noise. Once the noise is detected based on the calculated probability threshold, it is rescaled to match the mean value of the regular data, effectively reducing the impact of swell noise and enhancing the overall data quality. Supplementary Fig. 12 shows a gather before and after swell noise removal, showing cleaner signals.

**Predictive deconvolution.** To remove the ringing bubble pulse generated by the airgun source, two predictive deconvolution filters were computed for each sail line – one for the port source and one for the starboard source. These filters were calibrated using an autocorrelation analysis of the source wavelet to ensure optimal suppression of the ringing effect (3–75 Hz). Supplementary Fig. 13 shows an example of gather before and after the predictive deconvolution, showing a significant decrease in the low-frequency bubble pulse noise.

**Spherical divergence correction.** Seismic energy decays away from the source due to geometrical spreading. To address the amplitude decay caused by geometric spreading, a spherical divergence correction was applied using RMS (root-mean-square) velocities derived

from tomographic interval velocity, using the following formula:

$$A_{comp}(t) = A_{raw}(t) \cdot \left( \frac{v_{rms}(t) \cdot t}{v_0} \right)^2,$$

where, $A_{comp}$ and $A_{raw}$ represent the amplitudes before and after compensation, $v_{rms}$ is the root mean square (RMS) velocity, $v_0$ is the reference velocity, which is 1.5 km/s. $t$ is the two-travel travel time at zero offset.

**Linear dipping noise removal.** As there is a strong velocity contrast at the seafloor and the basaltic seafloor is rough, seafloor scattering and diffraction occur that travel in the water column with water velocity, leading to linear noise (e.g., Supplementary Fig. 12a). This linear dipping noise was removed using a dip filter in the shot domain. The method used isolated and suppressed unwanted energy along specific slopes (<1.8 km/s) in the frequency-wavenumber ($f$-$k$) domain, preserving primary reflections. Supplementary Fig. 14 shows a gather before and after the linear dipping noise removal.

**Suppression of energy travelling with low velocities.** Abnormal energy travelling with low velocities (1.5 km/s), such as side-scattering noise and mode-converted events, were identified and excluded in the common midpoint (CMP) gather domain using a $f$-$k$ (frequency-wavenumber) filtering technique. The CMP gathers were first corrected for normal moveout using the RMS velocity derived from 80% of the tomographic velocity. Consequently, the abnormal low-velocity signals appeared as downward-dipping components, which were then removed using a dip filter, followed by an inverse normal moveout correction. Supplementary Fig. 15 shows a gather before and after the suppression of energy travelling with low velocities.

**Frequency optimisation.** To improve signal clarity, a Butterworth bandpass filter with a frequency range of 3–35 Hz was applied. This filter was chosen for its smooth roll-off and minimal phase distortion, ensuring that primary seismic reflections remained intact within the optimised frequency band.

**Surgical muting.** Refracted and surface-related multiple energy were suppressed through a surgical mute, interactively applied based on amplitude and velocity characteristics. This step ensured that unwanted signals were excluded without compromising the integrity of the target reflections.

## 3D pre-stack depth migration (PSDM)

We applied a 3D Kirchhoff PSDM[61], employing a detailed workflow to ensure accurate subsurface imaging and optimal use of the dataset. Key aspects of the methodology are outlined below:

**Migration aperture.** To ensure comprehensive subsurface imaging, migration was performed with an Inline aperture of 6 km and a Xline aperture of 1 km. We systematically tested Xline apertures of 1 km, 2 km, and 4 km (Supplementary Fig. 16). Increasing the Xline aperture enhanced the sharpness of steeply dipping structures (i.e., lithosphere-asthenosphere boundary) but also amplified shallow migration artifacts (smiles) that disrupted imaging of the volcanic stratigraphy (Supplementary Fig. 16). However, a narrow aperture led to lateral smearing. To achieve an optimal balance between structural clarity and an objective interpretation, the 1-km aperture image was selected for our final interpretation while using the 4-km aperture image to avoid interpreting smeared image. This choice ensures sufficient data redundancy for coherent imaging while maintaining computational efficiency.

**Velocity model refinement.** The velocity model was iteratively refined through residual move-out (RMO) analysis applied to common-image gathers (CIGs), obtained after the application of PSDM. If the migration velocity is correct, reflection events in the CIG domain would become flat (i.e., horizontal in depth). Therefore, misaligned reflection events were systematically corrected by adjusting the velocity model, ensuring accurate alignment of events across all offsets. The updated model improved the consistency and focus of migrated reflections. Supplementary Fig. 17 shows a CIG before and after this correction.

**Muting strategy.** After migration, far offset arrivals were stretched in depth. An aggressive muting strategy was applied to the CIGs to enhance image quality. This process selectively retained near-offset data, minimising the impact of long-offset noise and artifacts. The mute was designed to make sure all the wide-angle energy associated with Layer 2A was excluded. The muting criteria were interactively determined based on velocity analysis and reflection continuity. Supplementary Fig. 18 shows an example of a surgical mute used on these data.

**Imaging.** A 3D Kirchhoff PSDM[61] was performed by tracing ray-paths through the refined velocity model to accurately position reflection events in their true subsurface locations. The migration process transforms the pre-stack seismic data into CIGs in the depth domain with high spatial resolution ($37.5 \, m \times 12.5 \, m \times 5 \, m$). These CIGs were summed to produce 3D PSDM image (Supplementary Movies 1 and 2). This method captured structural details, while preserving amplitude variations, critical for subsequent interpretation and analysis.

**Kirchhoff versus Reverse Time Migration (RTM).** One advantage of the Kirchhoff migration method is its computational efficiency. However, as it is based on ray theory, which does not allow it to accurately compute travel times and amplitudes in the case of high velocity contrasts, such as basaltic seafloor and melt bodies, its application may lead to migration artefacts such as smearing and migration smiles. Supplementary Fig. 16 shows that the smearing is more prominent when the aperture in the crossline direction is small (1 km), whereas the migration smiles in the upper part of the image are more pronounced when the aperture is large (4 km). On the other hand, reverse time migration (RTM)[62] is based on the two-way wave equation, which allows to accurately model wave propagation in the case of complex geology and high velocity contrasts, and hence RTM has the potential to produce a better quality image. However, it is computationally very expensive as it based on numerical modelling of the wave equation.

To assess the validity of our approach, we performed 2D PSDM using both Kirchhoff and RTM methods. By analysing our 3D PSDM image, we found that the smearing and migration smiles are more prominent in the Xline direction, possibly due to the coarser spatial sampling in the crossline direction (37.5 m), and therefore, we decided to test the two methods on a 2D profile acquired along the 3D box crossline direction (profile 2003), which was acquired using similar acquisition parameters. The results are shown in Supplementary Fig. 19. We find that the 2D Kirchhoff image (Supplementary Fig. 19a) shows weak layered reflections above the magma domain; these features are more pronounced in the 2D RTM image (Supplementary Fig. 19b), but to a comparable degree as in the 3D Kirchhoff image (Supplementary Fig. 16a), which gives us confidence that we could interpret the 3D PSDM image for structures above the magma domain.

**Criteria for the identification of different events in seismic profiles**
To systematically identify key geological features in seismic profiles, we took seismic traces extracted from Inline 261 as examples, focusing on Xline locations 2275, 2775, and 3350 (Supplementary Fig. 3). These profiles were examined using multiple seismic attributes, including

waveform characteristics, instantaneous amplitude, instantaneous phase, and time-wavenumber spectra. The classification of seismic events was based on signal continuity, and distinct amplitude and phase properties, allowing for the differentiation of seafloor, lava flow layers, melt sills injected into lava flow layering and the axial magma lens (AML) | lithosphere-asthenosphere boundaries (LAB) reflections.

**Seafloor identification.** The seafloor reflection is the most prominent feature in the seismic image, characterised by the strongest instantaneous amplitude response in the dataset; an instantaneous phase transition from negative to positive; and the highest wavenumber among identified events (Supplementary Fig. 3). These characteristics ensure a clear and unambiguous identification of the seafloor within the whole data volume.

**Lava flow layer identification.** Lava flow layers exhibit distinct seismic attributes that set them apart from deeper geological structures: an intermediate amplitude, lower than the seafloor and LAB, but still clearly discernible; a phase transition from negative to positive, similar to the seafloor; relatively high wavenumber, indicating a sharper and more abrupt change in subsurface properties (Supplementary Fig. 3). Their lateral continuity suggests that they generally initiate near the seafloor and gently dip towards the magma domain or rift zones, with the dip varying from nearly zero reaching up to 18° (Fig. 5). These features are essential in distinguishing lava flow layers from deeper seismic events, particularly when considering their spatial distribution and continuity.

**LAB identification.** The LAB is a crucial structural interface in our seismic image analysis, displaying as a moderately strong amplitude event, lower than the seafloor but still well-defined; a phase transition from positive to negative, contrasting with the seafloor and lava flow layers; a lower wavenumber compared to the overlying features, indicative of a broader-scale transition (Supplementary Fig. 3). These characteristics facilitate the identification of the LAB in the seismic sections, ensuring its proper distinction from overlying and underlying events.

**Injected melt sill identification.** Melt sills injected into the upper crust exhibit seismic properties similar to the melt sills in the magma domain but can be differentiated through their relative position to the identified LAB; comparable amplitude and phase characteristics, with a positive-to-negative phase transition (Supplementary Fig. 3); their spatial locations nearly parallel to lava flow layering (e.g., Fig. 7). By leveraging these distinguishing features, these upward and outward injected melt sills can be systematically identified and differentiated from the melt sills below the LAB in the magma domain, enhancing the geological interpretation of the seismic dataset.

**Travel-time tomography and full waveform inversion of 2D data**
The conventional seismic image shows the presence of an event, which is generally interpreted as Layer 2A event (Fig. 1a), but we do not observe any such event in the 3D PSDM image (Figs. 3–8). To understand the origin of this event, a high-resolution P-wave velocity was constructed beneath the seafloor, using a combination of travel time tomography and full waveform inversion (FWI) applied to a 2D long-offset multi-channel seismic (MCS) profile (2003) collected during the same survey (See Supplementary Fig. 1 for the location). For the 2D acquisition, the streamer length was increased to 12 km, and both sources were combined to create one larger source. This profile is nearly coincident with Xline 2448 in the 3D box.

**Initial velocity model.** The initial velocity model for travel time tomography was extracted from the velocity model used in the pre-

processing for 3D PSDM (Supplementary Fig. 10) but was resampled along the 2D profile. To ensure compatibility with the travel time inversion, the velocity model was extrapolated and interpolated (Supplementary Fig. 20a).

**Travel time tomography.** To construct a robust initial model for FWI, we first employed an iterative linearised travel time tomography approach[63] (Supplementary Fig. 20b). Travel times of first arrival were manually picked for every fourth shot gather, ensuring that at least one shot was located within the first Fresnel zone at the seafloor. Following the approach of Arnulf et al.[32], a uniform travel time uncertainty of 12 ms was assigned to all picked arrivals. In regions affected by shadow zones, first-arrival refractions were traced to the last reliably observed time, and reflections from the melt sill-LAB were picked at far offsets (e.g., Supplementary Figs. 21, 22). After seven iterations, the $\chi^2$ misfit was reduced to ~10% of its initial value (Supplementary Fig. 23a). The final inverted velocity model is shown in Supplementary Fig. 20b.

**Full waveform inversion (FWI).** After determining the final tomographic velocity model, FWI[64] was performed on the 2D dataset (Supplementary Figs. 21, 22). A careful time-windowing strategy was employed, preserving all refracted and reflected wave information. A Butterworth low-pass filter with a cut-off frequency of 3–6 Hz was applied to the seismic data before the inversion (e.g., Supplementary Figs. 21, 22). A source wavelet was extracted from the stacked zero-offset seafloor reflections. The L2 norm (true amplitude) objective function was used to quantify the discrepancy, and the nonlinear conjugate gradient method was used to iteratively update the model[64]. A total of 20 iterations were performed, progressively refining the model to enhance data fit (Supplementary Fig. 23b). The final inverted velocity model is shown in Supplementary Fig. 20c and the velocity anomaly in Supplementary Fig. 20d. A set of 1D velocity-depth profiles along profile 2003 is shown in Supplementary Fig. 8.

**Checkerboard test.** To assess the resolution of FWI, we performed checkerboard tests using three different checkerboard sizes, 1.5 km × 1 km, 1 km × 1 km and 0.5 km × 0.5 km with ±5% of sinusoidal perturbation added to the smoothed version of the final velocity model. We computed synthetic seismograms using these perturbed models and performed FWI starting from the smoothed velocity model, keeping all the parameters the same as the FWI of the real seismic data. The result of inversion is shown in Supplementary Fig. 24, which shows that velocity anomaly of the size 1 km × 1 km could be resolved down the base of the model and of the size 0.5 km × 0.5 km could be resolved down 1.5 km below the seafloor. As the Layer 2 A/2B boundary is laterally continuous, we can resolve this boundary within a vertical uncertainty of ±0.5 km.

**Porosity estimation**
To quantify variations in rock versus water content near the conventional Layer 2A/2B boundary, a differential effective medium (DEM) approach was employed[65] to establish velocity-porosity relationships under different inverse aspect ratio ($\alpha$) conditions (Supplementary Fig. 8b). The inverse aspect ratio is the ratio of the polar to the equatorial radius and is a measure of pore or crack shape. In our model, seawater properties were set to: P-wave velocity of 1.5 km/s and S-wave velocity of 0 km/s, with a density of $1.03 \times 10^3 \text{ kg/m}^3$. The surrounding basalt matrix was assigned a P-wave velocity of 6.8 km/s, an S-wave velocity of 3.8 km/s, and a density of $2.9 \times 10^3 \text{ kg/m}^3$. These parameters provided a physically consistent framework for evaluating porosity variations based on seismic velocity measurements. In the case of $\alpha = 5$[42], porosity near the conventional Layer 2A/2B boundary in FWI result decreased from 18% from seafloor to 4% at ~1.5 km depth (Supplementary Fig. 8).

## Data availability

MCS data for the 3-D cruise is available on the Marine Geoscience Data System at https://www.marine-geo.org/; the cruise information name is MGL1905, https://doi.org/10.7284/908292. Correspondence and requests for materials should be addressed to Satish C. Singh (singh@ipgp.fr).

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

## Acknowledgements

We thank the officers and crew of the R/V Marcus Langseth and supporting Office of Marine Operations at Lamont-Doherty Earth Observatory, Columbia University. We also thank the around-the-clock shipboard open-participation crew and onshore liaison: Axelle Cap, Brian Oiler, Massimo Bellucci, Matthews Griffiths, Michelle Lee, Morgane Goulain, Shelby Brandt, Tanner Eischen, Victoire Lucas, Samuel Mitchell and Annie Kell. Data-processing software was provided by Aspen Tech, Inc. (Paradigm 22), Bedford, Massachusetts, USA. Echos and Geodepth were used for the three-dimensional binning, stacking and prestack depth migration of the multichannel seismic data. Data presentation of the three-dimensional seismic volume was facilitated through visualization using Matlab. This project was funded through the National Science Foundation awards: OCE-1658021 (G.M.K.), OCE-1658199 (A.F.A.), and OCE-1658018 (A.J.H.). H.W. and W.X. were funded by Chinese Scholarship Council. S.C.S. was in part supported by ERC Advanced Grant MohoLAB 101141564. IPGP contribution number 4298.

## Author contributions

A.F.A. and A.J.H. collected the 3-D dataset onboard the R/V Marcus Langseth; 3D PSDM was performed by H.W. and 2D FWI was performed by W.X., under the supervision of S.C.S., H.C., and G.M.K. Interpretation of the datasets was undertaken by S.C.S., H.C., G.M.K. and H.W. and manuscript was written by S.C.S., G.M.K., H.C., and H.W. with input from A.F.A. and A.J.H.

## Competing interests

The authors declare no competing interests.
