## [Transparent Peer Review file · Nature Communications]

Oceanic upper crustal accretion by melt sill and lava flow interaction at Axial volcano

Corresponding Author: Professor Satish Singh

Version 0:

Reviewer comments:

Reviewer #1

(Remarks to the Author)

Wu et al present some very significant results from a detailed 3D seismic imaging experiment over a seamount at the Juan de Fuca Ridge. Their finding provide our first glimpse at the volcanic stratigraphy that comprises the volcanic pile of this hotspot seamount, and shows that these edifices are built from a complex magma feeder system. It establishes that even seamounts the size of Axial Volcano are more similar to Iceland or Hawaii than to a normal type of mid-ocean ridge. It's important to note that this type of seismic imaging is virtually impossible for subaerial volcanoes, since this study involves floating the seismic source and receivers in straight lines and close spacings directly over the edifice, and this can only be done at sea. I feel like the implications of this study will go well beyond the immediate hotspot-influenced ridge segment, since Axial Volcano is in my opinion and as shown here, quite analogous to many other basaltic central volcanoes.

I find the work supports the claims it makes. There are no significant methodological errors. The data processing is quite well done, as is typical for Singh and his group, and the seismic reflection results also supported by tomography and FWI (lines 498-529 and figures referenced therein).

I have several suggestions of a minor nature. Several places I feel the wording could be improved. An annotated manuscript is attached as it is much easier to see those suggestions there. I have two main suggestions. First that the authors avoid the use of the term "lava flows" as they note they are not imaging individual lava flows (lines 116-122). It's a bit of an open question what they are imaging, but it almost certainly is volcanic layering. Thus, my preference is to use "volcanic layering" or "lava stratigraphy" in lieu of lava flows. I realize that it is attractive to use the words melt sill and lava flow, as a sort of geological pair, but the imaging is probably melt sill packages and a kind of overall lava stratigraphy (not lava flows but packages of lava flows). Second, the figures - importantly the main figures - have different geographic references. These could be confusing to those not familiar with the jargon of the place names: Axial, CoAxial, NRZ, etc. I would suggest labeling the caldera, NRZ, SRZ, and Helium Basin on every figure that those appear. Aside from these minor edits and some wording, I have no other concerns.

Reviewer #2

(Remarks to the Author)

Review by William Chadwick

General comments on the manuscript:

This paper presents results from a 3D multi-channel seismic survey of Axial Seamount collected in 2019. It complements another paper by some of the same authors that was recently published in Nature (Kent et al., 2025) that focuses on the imaging of the lithosphere-aesthenosphere boundary (LAB), which Kent et al. interpreted as the upper surface of the "magma domain" beneath the volcano (a volume that includes stacked melt sills embedded in crystal mush). This paper represents a more detailed look at the shallow structure visible in the seismic imagery, both above and below that LAB boundary, by processing the seismic data using different methods, and focuses how the extrusive lava flow section interacts with injected sills from the magma domain.

One provocative interpretation is that there is a total lack of a sheeted dike complex beneath Axial Seamount, which the authors interpret as due to crustal assimilation from the magma domain (essentially melting upwards until it is in direct contact with the extrusive layer). The authors interpret that this is a product of Axial's (current) high magma supply, being the

location of a hotspot superimposed on a spreading ridge, and the implication is that this may be the case at other sites of plume-ridge interaction.

Another provocative interpretation related to the first is that the traditional interpretation of the seismically-defined Layer 2A/2B boundary as being the boundary between extrusive lavas above and sheeted dikes below does not apply to Axial (even though that "classical" view has been applied here in the past by other researchers). Instead, these authors interpret that what has been previously mis-interpreted as the "classical Layer 2A/2B boundary" here, is actually due to a steep velocity gradient at the transition from extrusive lava flows in the shallowest crust to the zone where melt sills have been injected into the lava flow layers from the magma domain, below a depth of about 1 km.

Overall, the paper is well-written, the figures are excellent, and the provocative interpretations will be of interest to a broad segment of the marine and earth science research community. I think it will stimulate a lot of discussion and debate, and perhaps re-evaluation of older datasets. I like that it challenges an old assumption that has been taken for granted for a long time, but state-of-the-art data show might not be true everywhere.

However, there is room for improvement. One pervasive problem is that the authors do not clearly distinguish between interpretation and fact in much of the manuscript. For example, what are identified as "lava layering", "injected sills", and "faults" in the seismic images are interpretation, and it should be made clearer early on what the criteria are for identifying (interpreting) these, and the differences between them. Many of my comments and suggested edits are related to where interpretations need to be more clearly identified as such, or statements qualified.

In addition, I find there are problems with some of the terminology used in the paper:

- Since the seismic images are not showing individual lava flows (see lines 127-130 in the ms), the text should be careful to not refer to individual lava flows in the seismic images! The authors should refer to the imaging of "lava flow layering" instead (which is an interpretation).

- The authors adopted the terminology of "magma domain" to describe the volume of melt and mush that exists within the volcano. However, in places they mix in the terms "magma reservoir" or "melt body", which to me is confusing (are they making a distinction? describing something different?). I think they should be consistent with their terminology and not use those other terms, because to me they convey slightly different meanings and they are not really interchangeable.

- The authors should not use the term "bedding planes", because that term refers to sedimentary rock structures, not lava flows. They should use "layers" instead.

Other important corrections that need to be made pertain to the age of the current caldera and the description of how dike propagate and how lavas are emplaced during the most recent eruptions (see specific comments below). Also, the very illuminating animation videos of the 3D seismic data in the Supplementary Materials are never cited or mentioned in the manuscript text! They should be! Finally, some of the referencing needs to be improved.

In sum, this study will be of interest to a wide audience and will make a strong contribution to Nature Communications. I recommend publication after moderate revision to address the comments and suggested edits below and in the annotated manuscript file (which has the figures removed to make the file size more manageable).

Specific comments (keyed to lines in the annotated manuscript):

Line 2: I think you need to add "Juan de Fuca Ridge, NE Pacific" to the end of the title so potential readers know that Axial is a submarine volcano on a spreading ridge.

Line 26: In the Kent et al. Nature paper, the term "magma domain" was used for the zone that includes both melt and mush, so I think that same terminology should be used in this paper, for consistency.

Line 29: How do you know the lava flows are "hydrous"? This seems like speculation.

Line 30: "bedding planes" is a term relevant to sedimentary rocks. "layers" is a better term for lava flows.

Lines 45-47: Is the second half of this sentence really a widely enough held view that it warrants mentioning here?

Line 54: I understand that you want to use precise seismological terminology ("the stacking of a retrograde refraction/reflection branch"), but this phrase does not mean anything to someone like me who is not a seismologist. Is there another way to say this that would be more meaningful to a non-seismologist? For example, I understand what a "high velocity gradient" is. Maybe just switch the order of the two descriptions, so that the "high velocity gradient" comes first in the sentence and the more precise description comes after, perhaps in parentheses?

Line 56: I find it confusing that there are 6 main figures, and then 8 additional "Extended Data Figures" in this manuscript file. Then there are an additional 17 Figures in the separate Supplementary Materials file. Looking at the guide to authors on the Nature Communications website, it says 10 display items are allowed per manuscript, and there is no mention of "Extended Data Figures". I wonder if some of the "Extended Data figures" should be moved to the Supplementary Materials file?

That said, I find Extended Data Figures 1-4 each provide important support for some of the main points in the paper, so I would encourage the authors to include them as main figures in the paper, not in the Supplementary Materials. If 10 figures are allowed, then the 6 main figures in the manuscript plus the first 4 "Extended Data figures" would be allowed.

Line 65: There are also these relevant references about Axial as a hotspot volcano:

Desonie, D. L., and R. A. Duncan (1990), The Cobb-Eickelberg seamount chain: Hotspot volcanism with mid-ocean ridge basalt affinity, *J. Geophys. Res.*, 95, 12697-12712

Chadwick, J., M. Perfit, I. Ridley, I. Jonasson, G. Kamenov, W. W. Chadwick, Jr., R. Embley, P. Le Roux, and M. Smith

(2005), Magmatic effects of the Cobb Hotspot on the Juan de Fuca Ridge, *Journal of Geophysical Research: Solid Earth*, 110, B03101, doi:10.1029/2003JB002767.

Line 66: There is also this relevant reference about the hydrothermal vents at Axial:

Butterfield, D. A., G. J. Massoth, R. E. McDuff, J. E. Lupton, and M. D. Lilley (1990), Geochemistry of hydrothermal fluids from Axial Seamount hydrothermal emissions study vent field, Juan de Fuca Ridge: Subfloor boiling and subsequent fluid-rock interaction, *J. Geophys. Res.*, 95(B8), 12895-12922

Line 70: There is also this relevant reference about the geology and structure of Axial:

Embley, R. W., K. M. Murphy, and C. G. Fox (1990), High resolution studies of the summit of Axial Volcano, *J. Geophys. Res.*, 95, 12785-12812

Lines 71-74: This correction is very important. The 31 kyr age of the caldera reported in Clague et al. 2013 is now thought to be erroneous (by Clague). Their current interpretation of the age of the caldera is that it is 1259 ± 119 yrs BP (where BP means before 1950, as 14C ages are commonly reported, so that is 691 CE ± 119 yrs). These revised dating results were reported in this 2019 AGU abstract:

Clague, D. A., R. A. Portner, J. B. Paduan, M. Le Saout, and B. M. Dreyer (2019), Formation of the summit caldera at Axial Seamount. Abstract presented at 2019 Fall Meeting, AGU, San Francisco, CA, 9-13 Dec.

... and will be published in a forthcoming paper that is currently in revision:

Paduan, J. B., D. A. Clague, D. W. Caress, R. Portner, M. Le Saout, and B. Dreyer (in revision), Voluminous Inflated Lobate Flows on the Distal Rift Zones Synchronous with Caldera Formation at Axial Seamount, Juan de Fuca Spreading Ridge, *Geochemistry, Geophysics, Geosystems*

However, the age for the caldera reported in the AGU abstract is slightly different than their current preferred age, so it may be best to cite "D. Clague, personal communication, 2025" in addition to the AGU abstract, and perhaps the paper in revision, if that is allowable.

Line 77: Reference #23 is not correct. It should be this one:

Kelley, D. S., J. R. Delaney, and S. K. Juniper (2014), Establishing a new era of submarine volcanic observatories: Cabling Axial Seamount and the Endeavour Segment of the Juan de Fuca Ridge, *Mar. Geol.*, 352, 426-450, doi:10.1016/j.margeo.2014.03.010.

Also, I think this sentence should be moved to the end of the previous paragraph, since the rest of this new paragraph is all about seismic imaging at Axial.

Line 92: Need to briefly say what "magma domain" means in this context...

Lines 93-94: I think it's important to state that you have processed the same dataset as presented in Kent et al. (2025) in a different way, to show how this paper relates to that one, and how it is different from it.

Line 132: I don't think change in the petrology would have any effect on their impedance contrast, but the physical properties of individual flows related to their morphology and emplacement history could.

Line 139: They aren't really "melt bodies". It is more of a boundary, right?

Line 161: I would think disruption by faulting & block rotation, etc, during previous caldera collapse events would be the main one.

Line 171: Should reference Chadwick et al. (2016), and Clague et al. (2017) here (refs #19 & #20).

Lines 172-176: The wording of this sentence is wrong. Dikes propagate laterally 10s of km down the rift zones and erupt lava from fissures locally (where the dike reaches the surface). The lava flows themselves do not flow for distances up to 30 km.

Lines 176-178: This sentence is also incorrect. The lava flows erupted in 1998, 2011, and 2015 are all thinner inside the caldera (generally 3-5 m near the eruptive fissures and up to 15 m downslope - but still inside the caldera), where fluid sheet flows were erupted. Lava flows are much thicker on the distal parts of the rift zones (well outside the caldera), where they form large mounds of pillow lava up to 137 m thick in 2011, and 128 m thick in 2016!

They don't "thin downslope" – in fact they usually do the opposite. They do generally "dip away from the caldera and rift zones" just because that is the slope of the underlying topography.

Lines 184-187: I think you should combine 2 & 3 here (as possible mechanisms that could cause subsidence), and just talk about (infrequent) large volume eruptions that cause caldera collapse (like the one described on the distal SRZ by Clague, or on the scale of the Kilauea eruption in 2018). That scale of eruption is the only kind that would cause significant enough changes in the shallow crust to affect the structure you see in the seismic imagery. Just moving magma from deep to shallow depth during a small-volume eruption would not do it. Also, I think the subsidence has to be a long-term process, NOT

something associated with the relatively small historical eruptions like in 1998, 2011, and 2015. Those events did not cause any significant permanent subsidence. For example, the observed vertical displacement of the seafloor (inflation/deflation) over the recent eruption cycles is just a few meters, and is mostly elastic and reversible.

Lines 188-207: This paragraph strikes me as overinterpretation. I'm very skeptical that the apparent 2D shape of melt sills beneath the caldera reflects magma withdrawal from individual (small) eruptions. I would omit this.

Lines 217-218: The "funnels" are not melt bodies. The "funnels" are the upper boundary or surface of the "magma domain".

Line 223: How do you know they are "altered"?

Line 224: How do you know they have a "water-rich mineralogy". What is the evidence for that? This and the remainder of this paragraph read like unconstrained speculation, and these sentences lack references to substantiate these claims. I think this paragraph needs to be re-written as a hypothesis, rather than stating these things as fact.

Line 228: Is there evidence for this (the alteration of magma compositions consistent with assimilation at Axial)?

Line 229: Does the chemistry of the andesite ridge lavas on the NRZ of Axial support his hypothesis? Andesite can form by fractionation alone, so its presence (by itself) is not evidence that assimilation is happening.

Line 266: This needs to be phrased as a hypothesis or interpretation, not fact.

Lines 279-281: Define what FWI is in the captions to Extended Data Figs. 6 & 7. The captions should also explain that they pertain to just one 2D profile (& indicate which Xline).

Lines 376-377: Methods section: These dimensions for Inline vs Xline are switched compared to what is said in lines 329 & 330. Is this correct? Or is this backwards?

Line 755: Figure 1 caption: The subduction plate boundary is not marked. Don't need this text anyway...

Line 783: Figure 4 caption: Where are the 6 locations? Need to add arrows to specifically show them so the reader isn't left guessing.

Line 801: Figure 5 caption: This phrase talking about rotation is potentially confusing, because it is the dip direction that is rotated $\sim 90^\circ$, not the dip angle (magnitude).

Lines 814-816: Figure 6 caption: What is the dark blue with the light blue lines on the left? Needs an explanation. Why would the left side be "less evolved"? I would think it might be more evolved, since the magma supply is waning, there is more cooling, so more opportunity for fractionation.

Line 851: Extended Data Figure 5: I'm very skeptical of what this is supposedly showing...

Line 859: Extended Data Figure 6: The caption should say something about what is shown by this figure and why it is included in the paper. What interpretation does it support?

Line 869: Extended Data Figure 7: specify which profile this is.

Line 877: Extended Data Figure 8: what is the "fold number"?

Reviewer #3

(Remarks to the Author)

SUMMARY

The paper uses depth migrated seismic images from the Juan de Fuca Ridge to provide a detailed model of crust layering. Authors find that lava flows can be identified on the sections, but the expected sheeted dykes are not identified. Since very few (if none) detailed seismic images of crust creation (if none) exist, the paper is a commendable contribution.

However, Kirchhoff depth migration, which is the migration algorithm used in the paper, is known to create artifacts in areas with large velocity contrasts, such as spreading ridges. The seismic sections presented seem to be somewhat smeared

and also contain migration smiles. This might influence the interpretation, and mask important details, in particular since a conclusion is that sheeted dykes are absent.

I recommend that the PSDM is redone with a more appropriate algorithm, or that at least a test is performed on a sub volume of the data.

3D PSDM

The PSDM pre-processing and noise removal flow seem to be adequate. However, the quality of the depth sections shown in the paper seems to be quite low. Although the wavelengths are large due to the crust velocity and low bandwidth of the data, the depth images seem to be too smeared. Some of the sections show visible smiles.

The velocity model contains large velocity contrasts due to the high velocities in the crust. The PSDM is using a Kirchhoff algorithm, which is based on ray theory. It is well known that Kirchhoff migration does not handle large velocity contrasts well and leads to smearing and migration smiles due to the limitations of ray theory.

Use of Kirchhoff migration will also probably affect the velocity analysis.

Redoing the PSDM with with reverse-time migration or even a one way wave based shot migration would probably give a better result. At least a test of a part of the data volume could be performed to investigate the sensitivity due to choice of migration algorithm.

2D FWI

The 2D Full Waveform Inversion workflow is adequate and the results seem to be sensible. Also the the residuals are substantially reduced from first to last iteration confirming convergence.

However, although residuals are small, resolution might still suffer. Some kind of resolution analysis would be good for verifying the quality

Version 1:

Reviewer comments:

Reviewer #1

(Remarks to the Author)

Review of revised "Oceanic upper crustal accretion by melt sill and lava flow interaction at Axial volcano"
The manuscript presents an interesting view of the crustal architecture at Axial, a type section of hotspot-influenced spreading ridge, and provocative interpretation of the magma delivery system that differs from the standard layered model. The key new contribution is the imaging of layering within the volcanics. This is new and is significant to the volcanology community. The manuscript has been revised addressing reviewers' comments. I find the sections on Magma Domain, Subsidence, and Hydrated-Dehydrated Boundary are very well supported. The section on Assimilation is a bit speculative and it is didn't see the point of a long-ish discussion of this topic given the previous comments by reviewers 1 and 2. This is the one area where I thought their conclusions were not strongly supported by the data. The new explanation of the trade-offs with the Kirchhoff migration seem to justify it well, and although reviewer 3 is technically correct that RTM could produce cleaner imaging, I find the images presented are clear enough to interpret easily. I find the methodology to be sound and reported in sufficient detail.

Lines 51-53: I agree that there is an unnecessary amount of jargon/detail here about "stacking a retrograde refraction/reflection branch" followed by Figure 1a which does not enlighten the reader about branches but does show layer 2A. Most people aside from seismologists would probably be happier with the layer 2A event being a reflection produced in a high velocity gradient zone.

Line 105: I was expecting a figure that showed the LAB geometry, and I feel like a better choice could be made than Figure 2.

Line 159: I'm not sure what "baseline" lava flow layers are, or how they differ from other lava flow layers.

Line 184: It's unclear how "lava flow units" are defined, but I can see that the volcanic pile does thicken overall as toward the rift zones and magma domain.

Reviewer #2

(Remarks to the Author)

Review of revised manuscript by William Chadwick

General comments on the manuscript:

This is my second review of this manuscript, which has been substantially revised. In general, the revisions have addressed most of my previous comments and concerns. However, the paper would benefit from another round of minor revision before publication. I have made suggested edits and some additional comments in an annotated manuscript Word file (with the figures removed to reduce the file size). A few of the more significant comments are included below.

Specific comments (keyed to lines in the annotated manuscript):

Line 100: Descriptions of what is shown in the Movies needs to be added to the Supplementary Materials file. There are currently no descriptions or explanations at all. For example, what is the colored surface that is added below the seafloor in Movies #3, #4, and #5?

Line 202: This sort of geometry is observed on land in Iceland (lava flows exposed on the east and west coasts dip toward the spreading plate boundary). You should highlight this similarity more since it seems very relevant. For example, this paper discusses the same mechanisms of spreading and subsidence leading to "inward dipping" lava flow stratigraphy:

Daignieres, M., V. Courtillot, R. Bayer, and P. Tapponnier (1975), A model for the evolution of the axial zone of mid-ocean ridges as suggested by Icelandic tectonics, *Earth Planet. Sci. Let.*, 26, 222–232.

The rest of this sentence I don't understand. What difference does it make whether a MOR is subaerial or submarine in regards to this volcano-tectonic process? And the seaward dipping reflections (SDRs) on continental margins seem like a completely different thing and NOT relevant to this discussion, so I would omit that.

The similarities between the cross-axis structure in Iceland and the inward-dipping lava flows you see at Axial deserve a more coherent discussion here.

Reviewer #3

(Remarks to the Author)

My comments on the full waveform inversion has been taken into account, a resolution test has been performed and adequately discussed. The issue with migration artifacts still remains, but the authors discussion has clarified the issue, and it is not likely that main results of the manuscript will be significantly changed with a more adequate migration algorithm.

REVIEWER COMMENTS

Reviewer #1 (Remarks to the Author):

Wu et al present some very significant results from a detailed 3D seismic imaging experiment over a seamount at the Juan de Fuca Ridge. Their findings provide our first glimpse at the volcanic stratigraphy that comprises the volcanic pile of this hotspot seamount, and shows that these edifices are built from a complex magma feeder system. It establishes that even seamounts the size of Axial Volcano are more similar to Iceland or Hawaii than to a normal type of mid-ocean ridge. It's important to note that this type of seismic imaging is virtually impossible for subaerial volcanoes, since this study involves floating the seismic source and receivers in straight lines and close spacings directly over the edifice, and this can only be done at sea. I feel like the implications of this study will go well beyond the immediate hotspot-influenced ridge segment, since Axial Volcano is in my opinion and as shown here, quite analogous to many other basaltic central volcanoes.

I find the work supports the claims it makes. There are no significant methodological errors. The data processing is quite well done, as is typical for Singh and his group, and the seismic reflection results also supported by tomography and FWI (lines 498-529 and figures referenced therein).

I have several suggestions of a minor nature. Several places I feel the wording could be improved.

The wording has been improved as suggested by this reviewer and also in more detail by reviewer 2.

An annotated manuscript is attached as it is much easier to see those suggestions there.

We thank the reviewer for providing an annotated file. We have addressed most of the comments pointed out by the reviewer.

I have two main suggestions. First that the authors avoid the use of the term "lava flows" as they note they are not imaging individual lava flows (lines 116-122). It's a bit of an open question what they are imaging, but it almost certainly is volcanic layering. Thus, my preference is to use "volcanic layering" or "lava stratigraphy" in lieu of lava flows. I realize that it is attractive to use the words melt sill and lava flow, as a sort of geological pair, but the imaging is probably melt sill packages and a kind of overall lava stratigraphy (not lava flows but packages of lava flows).

Considering the comments of this reviewer and reviewer 2, we have decided to use the term 'lava flow layering' everywhere.

We refer to melt sills when melt is likely to be in molten state and frozen sills when it has been solidified. We have shown in Supplementary Figure 3 how to distinguish them and mention this early on in the discussion, soon after the image is introduced.

Second, the figures - importantly the main figures - have different geographic references. These could be confusing to those not familiar with the jargon of the place names: Axial, CoAxial, NRZ, etc. I would suggest labeling the caldera, NRZ, SRZ, and Helium Basin on every figure that those appear.

As suggested by the reviewer, all the figures have been re-labeled.

Aside from these minor edits and some wording, I have no other concerns.

Reviewer #2 (Remarks to the Author):

Review by William Chadwick

General comments on the manuscript:

This paper presents results from a 3D multi-channel seismic survey of Axial Seamount collected in 2019. It complements another paper by some of the same authors that was recently published in Nature (Kent et al., 2025) that focuses on the imaging of the lithosphere-aesthenosphere boundary (LAB), which Kent et al. interpreted as the upper surface of the “magma domain” beneath the volcano (a volume that includes stacked melt sills embedded in crystal mush). This paper represents a more detailed look at the shallow structure visible in the seismic imagery, both above and below that LAB boundary, by processing the seismic data using different methods, and focuses how the extrusive lava flow section interacts with injected sills from the magma domain.

One provocative interpretation is that there is a total lack of a sheeted dike complex beneath Axial Seamount, which the authors interpret as due to crustal assimilation from the magma domain (essentially melting upwards until it is in direct contact with the extrusive layer). The authors interpret that this is a product of Axial’s (current) high magma supply, being the location of a hotspot superimposed on a spreading ridge, and the implication is that this may be the case at other sites of plume-ridge interaction.

Another provocative interpretation related to the first is that the traditional interpretation of the seismically-defined Layer 2A/2B boundary as being the boundary between extrusive lavas above and sheeted dikes below does not apply to Axial (even though that “classical” view has been applied here in the past by other researchers). Instead, these authors interpret that what has been previously mis-interpreted as the “classical Layer 2A/2B boundary” here, is actually due to a steep velocity gradient at the transition from extrusive lava flows in the shallowest crust to the zone where melt sills have been injected into the lava flow layers from the magma domain, below a depth of about 1 km.

Overall, the paper is well-written, the figures are excellent, and the provocative interpretations will be of interest to a broad segment of the marine and earth science research community. I think it will stimulate a lot of discussion and debate, and perhaps re-evaluation of older datasets. I like that it challenges an old assumption that has been taken for granted for a long time, but state-of-the-art data show might not be true everywhere. However, there is room for improvement. One pervasive problem is that the authors do not clearly distinguish between interpretation and fact in much of the manuscript. For example, what are identified as “lava layering”, “injected sills”, and “faults” in the seismic images are interpretation, and it should be made clearer early on what the criteria are for identifying (interpreting) these, and the differences between them. Many of my comments and suggested edits are related to where interpretations need to be more clearly identified as such, or statements qualified.

We have replaced lava flows with lava flow layering as suggested by the reviewer. We have defined injected sills and faults in the text.

In addition, I find there are problems with some of the terminology used in the paper:

- Since the seismic images are not showing individual lava flows (see lines 127-130 in the ms), the text should be careful to not refer to individual lava flows in the seismic images! The authors should refer to the imaging of “lava flow layering” instead (which is an interpretation).

Sure, we have used lava flow layering.

- The authors adopted the terminology of “magma domain” to describe the volume of melt

and mush that exists within the volcano. However, in places they mix in the terms “magma reservoir” or “melt body”, which to me is confusing (are they making a distinction? describing something different?). I think they should be consistent with their terminology and not use those other terms, because to me they convey slightly different meanings and they are not really interchangeable.

Indeed we use magma domain to describe the volume of melt and mush located beneath a top boundary (strong seismic reflection) that we interpret to represent the LAB (Kent et al., 2025), but there are high reflectivity zones outside of this magma domain, e.g. around the Co-Axial region, and north of the caldera, which could not be included in the Magma domain as we do not know if they are connected with the magma domain identified by Kent et al. Therefore, we have used the word ‘melt sill’ for some of these features, but removed ‘magma reservoir’.

- The authors should not use the term “bedding planes”, because that term refers to sedimentary rock structures, not lava flows. They should use “layers” instead.

Thanks to the reviewer for this remark, we have now replaced ‘bedding planes’ by ‘layers’.

Other important corrections that need to be made pertain to the age of the current caldera and the description of how dike propagate and how lavas are emplaced during the most recent eruptions (see specific comments below).

See comments below

Also, the very illuminating animation videos of the 3D seismic data in the Supplementary Materials are never cited or mentioned in the manuscript text! They should be! Finally, some of the referencing needs to be improved.

Thanks to the reviewer for pointing this out, we have now referenced all the figures and movies at their appropriate places in the main text and in the Methods.

In sum, this study will be of interest to a wide audience and will make a strong contribution to Nature Communications. I recommend publication after moderate revision to address the comments and suggested edits below and in the annotated manuscript file (which has the figures removed to make the file size more manageable).

Specific comments (keyed to lines in the annotated manuscript):

Line 2: I think you need to add “Juan de Fuca Ridge, NE Pacific” to the end of the title so potential readers know that Axial is a submarine volcano on a spreading ridge.

The Nature Communications requires the title to be less than 15 words; we already had 14 words and hence cannot add extra words as suggested by the reviewer. However, if the editor allows, we would be happy to include these words.

Line 26: In the Kent et al. Nature paper, the term “magma domain” was used for the zone that includes both melt and mush, so I think that same terminology should be used in this paper, for consistency.

Done as suggested

Line 29: How do you know the lava flows are “hydrous”? This seems like speculation.

The word 'hydrous' is used to suggest the presence of water in pore spaces. For a basaltic rock, the P-wave velocity is ~6.4 km/s with zero porosity, which decreases to 2.5 km/s for a porosity of 40%, with pore spaces filled with water. The hydrated and de-hydrated definition is based on full waveform inversion results. However, we have deleted the word 'hydrous' from the abstract.

Line 30: "bedding planes" is a term relevant to sedimentary rocks. "layers" is a better term for lava flows.

Modified as suggested by the reviewer.

Lines 45-47: Is the second half of this sentence really a widely enough held view that it warrants mentioning here?

Yes, this alternate interpretation has been frequently invoked. Since we shall be interpreting the layer L2A/2B boundary as a hydration/dehydration boundary, it is important to mention this early on.

Line 54: I understand that you want to use precise seismological terminology ("the stacking of a retrograde refraction/reflection branch"), but this phrase does not mean anything to someone like me who is not a seismologist. Is there another way to say this that would be more meaningful to a non-seismologist? For example, I understand what a "high velocity gradient" is. Maybe just switch the order of the two descriptions, so that the "high velocity gradient" comes first in the sentence and the more precise description comes after, perhaps in parentheses?

We agree with the reviewer and have re-written the text to clarify the point, brought the gradient information first before using seismic jargon.

Line 56: I find it confusing that there are 6 main figures, and then 8 additional "Extended Data Figures" in this manuscript file. Then there are an additional 17 Figures in the separate Supplementary Materials file. Looking at the guide to authors on the Nature Communications website, it says 10 display items are allowed per manuscript, and there is no mention of "Extended Data Figures". I wonder if some of the "Extended Data figures" should be moved to the Supplementary Materials file?

We are sorry about this, but this has been due to earlier submission where Extended Data was allowed.

That said, I find Extended Data Figures 1-4 each provide important support for some of the main points in the paper, so I would encourage the authors to include them as main figures in the paper, not in the Supplementary Materials. If 10 figures are allowed, then the 6 main figures in the manuscript plus the first 4 "Extended Data figures" would be allowed.

Thanks to the reviewer for the suggestion, we have moved Extended Data Figures 1-3, in the main paper, as Figures 1, 5 and 6. We also have brought Figure 10 (new) from previous Supplementary Figure 7, as this figure is important. We feel that Extended Data Figure 4 is very technical and hence moved to Supplementary Figure 3. The rest of the Extended Data figures have been moved to the supplementary material.

Line 65: There are also these relevant references about Axial as a hotspot volcano: Desonie, D. L., and R. A. Duncan (1990), The Cobb-Eickelberg seamount chain: Hotspot volcanism with mid-ocean ridge basalt affinity, J. Geophys. Res., 95, 12697-12712

Chadwick, J., M. Perfit, I. Ridley, I. Jonasson, G. Kamenov, W. W. Chadwick, Jr., R. Embley,

P. Le Roux, and M. Smith (2005), Magmatic effects of the Cobb Hotspot on the Juan de Fuca Ridge, *Journal of Geophysical Research: Solid Earth*, 110, B03101, doi:10.1029/2003JB002767.

We thank the reviewer for pointing out these references, which we have cited and included in the paper.

Line 66: There is also this relevant reference about the hydrothermal vents at Axial: Butterfield, D. A., G. J. Massoth, R. E. McDuff, J. E. Lupton, and M. D. Lilley (1990), Geochemistry of hydrothermal fluids from Axial Seamount hydrothermal emissions study vent field, Juan de Fuca Ridge: Subfloor boiling and subsequent fluid-rock interaction, *J. Geophys. Res.*, 95(B8), 12895-12922

Once again, we thank the reviewer for pointing out this paper, which we have cited and included in the reference list.

Line 70: There is also this relevant reference about the geology and structure of Axial: Embley, R. W., K. M. Murphy, and C. G. Fox (1990), High resolution studies of the summit of Axial Volcano, *J. Geophys. Res.*, 95, 12785-12812

We have cited this paper and included it in the reference list.

Lines 71-74: This correction is very important. The 31 kyr age of the caldera reported in Clague et al. 2013 is now thought to be erroneous (by Clague). Their current interpretation of the age of the caldera is that it is 1259 ± 119 yrs BP (where BP means before 1950, as ^{14}C ages are commonly reported, so that is $691 \text{ CE} \pm 119$ yrs). These revised dating results were reported in this 2019 AGU abstract:

Clague, D. A., R. A. Portner, J. B. Paduan, M. Le Saout, and B. M. Dreyer (2019), Formation of the summit caldera at Axial Seamount. Abstract presented at 2019 Fall Meeting, AGU, San Francisco, CA, 9-13 Dec.

... and will be published in a forthcoming paper that is currently in revision: Paduan, J. B., D. A. Clague, D. W. Caress, R. Portner, M. Le Saout, and B. Dreyer (in revision), Voluminous Inflated Lobate Flows on the Distal Rift Zones Synchronous with Caldera Formation at Axial Seamount, Juan de Fuca Spreading Ridge, *Geochemistry, Geophysics, Geosystems*

However, the age for the caldera reported in the AGU abstract is slightly different than their current preferred age, so it may be best to cite "D. Clague, personal communication, 2025" in addition to the AGU abstract, and perhaps the paper in revision, if that is allowable.

We have modified the age of the caldera formation and have cited the above AGU abstract. We solicited Clague for further information and consent to use personal communication but he preferred us to cite the AGU abstract.

Line 77: Reference #23 is not correct. It should be this one: Kelley, D. S., J. R. Delaney, and S. K. Juniper (2014), Establishing a new era of submarine volcanic observatories: Cabling Axial Seamount and the Endeavour Segment of the Juan de Fuca Ridge, *Mar. Geol.*, 352, 426-450, doi:10.1016/j.margeo.2014.03.010.

Thanks for pointing out this error; the reference has been modified.

Also, I think this sentence should be moved to the end of the previous paragraph, since the rest of this new paragraph is all about seismic imaging at Axial.

Moved at the end of previous paragraph as suggested.

Line 92: Need to briefly say what "magma domain" means in this context...

A sentence has been added to explain the 'magma domain' as suggested by the reviewer.

Lines 93-94: I think it's important to state that you have processed the same dataset as presented in Kent et al. (2025) in a different way, to show how this paper relates to that one, and how it is different from it.

We have added a sentence saying that we use the same dataset as Kent et al. (2025), a paper which presented post-stack time migration results. For this study we processed the data using pre-stack depth migration algorithm, and our results allow to map structures in depth instead of time, and also show features that were not present in the Kent et al. (2025) image.

Line 132: I don't think change in the petrology would have any effect on their impedance contrast, but the physical properties of individual flows related to their morphology and emplacement history could.

This statement has been deleted.

Line 139: They aren't really "melt bodies". It is more of a boundary, right?

We have replaced melt bodies with boundaries.

Line 161: I would think disruption by faulting & block rotation, etc, during previous caldera collapse events would be the main one.

We have slightly modified the text and have emphasised that faulting and block rotation might be main causes, though we have no further insight on that.

Line 171: Should reference Chadwick et al. (2016), and Clague et al. (2017) here (refs #19 & #20).

We have included the above two references here.

Lines 172-176: The wording of this sentence is wrong. Dikes propagate laterally 10s of km down the rift zones and erupt lava from fissures locally (where the dike reaches the surface). The lava flows themselves do not flow for distances up to 30 km.

We agree with the reviewer and have modified the sentence to reflect the above points.

Lines 176-178: This sentence is also incorrect. The lava flows erupted in 1998, 2011, and 2015 are all thinner inside the caldera (generally 3-5 m near the eruptive fissures and up to 15 m downslope - but still inside the caldera), where fluid sheet flows were erupted. Lava flows are much thicker on the distal parts of the rift zones (well outside the caldera), where they form large mounds of pillow lava up to 137 m thick in 2011, and 128 m thick in 2016!

Thanks to the reviewer for pointing out these aspects. We have modified the text to take above comments into account.

They don't "thin downslope" – in fact they usually do the opposite. They do generally "dip away from the caldera and rift zones" just because that is the slope of the underlying topography.

We have deleted this statement.

Lines 184-187: I think you should combine 2 & 3 here (as possible mechanisms that could cause subsidence), and just talk about (infrequent) large volume eruptions that cause caldera collapse (like the one described on the distal SRZ by Clague, or on the scale of the Kilauea eruption in 2018). That scale of eruption is the only kind that would cause significant enough changes in the shallow crust to affect the structure you see in the seismic imagery. Just moving magma from deep to shallow depth during a small-volume eruption would not do it. Also, I think the subsidence has to be a long-term process, NOT something associated with the relatively small historical eruptions like in 1998, 2011, and 2015. Those events did not cause any significant permanent subsidence. For example, the observed vertical displacement of the seafloor (inflation/deflation) over the recent eruption cycles is just a few meters, and is mostly elastic and reversible.

We have combined the 2nd and 3rd causes into one and have rewritten the text to take the reviewer's comments into account.

Lines 188-207: This paragraph strikes me as overinterpretation. I'm very skeptical that the apparent 2D shape of melt sills beneath the caldera reflects magma withdrawal from individual (small) eruptions. I would omit this.

We have removed the second part of this paragraph as suggested by the reviewer.

Lines 217-218: The "funnels" are not melt bodies. The "funnels" are the upper boundary or surface of the "magma domain".

The funnels could have a 3D morphology, but in the present context, they represent the upper boundary of the magma domain (Kent et al., 2025). We have modified the text to take the reviewer's remark into account.

Line 223: How do you know they are "altered"?

We have deleted the word 'altered' and have rewritten the text to clarify the points.

Line 224: How do you know they have a "water-rich mineralogy". What is the evidence for that? This and the remainder of this paragraph read like unconstrained speculation, and these sentences lack references to substantiate these claims. I think this paragraph needs to be re-written as a hypothesis, rather than stating these things as fact.

We have deleted the phrase 'water-rich mineralogy' and rewritten the text, included several references to support the arguments.

Line 228: Is there evidence for this (the alteration of magma compositions consistent with assimilation at Axial)?

We do not have any direct evidence of assimilation in lava flows on the seafloor, which we clearly state in the text. Dr Philipp Ruprecht at the University of Nevada Reno is currently analysing some of the rock sample of Clague et al (2018) using the methodology of France et al. (2021) to verify the presence of assimilation.

Line 229: Does the chemistry of the andesite ridge lavas on the NRZ of Axial support his hypothesis? Andesite can form by fractionation alone, so its presence (by itself) is not evidence that assimilation is happening.

Yes, we contacted Mike Perfit of University of Florida, who confirmed the presence of assimilation. We cite their AGU abstract.

Line 266: This needs to be phrased as a hypothesis or interpretation, not fact.

We agree with the reviewer, we have rewritten the text as a possibility, not as a fact.

Lines 279-281: Define what FWI is in the captions to Extended Data Figs. 6 & 7. The captions should also explain that they pertain to just one 2D profile (& indicate which Xline).

We have defined FWI in the Figure Caption of this Extended Data Figures 6 (which is now Fig. 10) and also mention the line numbers, and also marked on Supplementry Fig 19-24.

Lines 376-377: Methods section: These dimensions for Inline vs Xline are switched compared to what is said in lines 329 & 330. is this correct? Or is this backwards?

Thanks for pointing this out, we have now corrected the numbers.

Line 755: Figure 1 caption: The subduction plate boundary is not marked. Don't need this text anyway...

We have removed the subduction plate boundary from the Figure caption.

Line 783: Figure 4 caption: Where are the 6 locations? Need to add arrows to specifically show them so the reader isn't left guessing.

We have included arrows to link the two figures.

Line 801: Figure 5 caption: This phrase talking about rotation is potentially confusing, because it is the dip direction that is rotated $\sim 90^\circ$, not the dip angle (magnitude).

This statement has been deleted.

Lines 814-816: Figure 6 caption: What is the dark blue with the light blue lines on the left? Needs an explanation. Why would the left side be "less evolved"? I would think it might be more evolved, since the magma supply is waning, there is more cooling, so more opportunity for fractionation.

We agree with the reviewer and have removed 'more' and 'less' as we do not know if they are more or less evolved, but have kept 'evolved', which is certain to be the case along this boundary.

Line 851: Extended Data Figure 5: I'm very skeptical of what this is supposedly showing...

As we have deleted the text based on this figure, we have removed this figure.

Line 859: Extended Data Figure 6: The caption should say something about what is shown by this figure and why it is included in the paper. What interpretation does it support?

The figure caption has been extended and interpreted to reflect the purpose of this figure. This figure has been moved to Fig 10.

Line 869: Extended Data Figure 7: specify which profile this is.

We have extended the Figure caption to explain the figure and also cited in the paper.

Line 877: Extended Data Figure 8: what is the "fold number"?

We have defined the fold number both in Figure caption and in the Methods.

Reviewer #3 (Remarks to the Author):

SUMMARY

The paper use depth migrated seismic images from the Juan de Fuca Ridge to provide a detailed model of crust layering. Authors find that lava flows can be identified on the sections, but the expected sheeted dykes are not identified. Since very few (if none) detailed seismic images of crust creation (if none) exists, the paper is a commendable contribution.

However, Kirchhoff depth migration, which is the migration algorithm used in the paper, is known to create artifacts in areas with large velocity contrasts, such as spreading ridges. The seismic sections presented seem to be somewhat smeared and also contain migration smiles. This might influence the interpretation, and mask important details, in particular since a conclusion is that sheeted dykes are absent.

I recommend that the PSDM is redone with a more appropriate algorithm, or that at least a test is performed on a sub volume of the data.

3D PSDM

The PSDM pre-processing and noise removal flow seem to be adequate. However, the quality of the depth sections shown in the paper seems to be quite low. Although the wavelengths are large due to the crust velocity and low bandwidth of the data, the depth images seem to be too smeared. Some of the sections show visible smiles.

The velocity model contains large velocity contrasts due to the high velocities in the crust. The PSDM is using a Kirchhoff algorithm, which is based on ray theory. It is well known that Kirchhoff migration does not handle large velocity contrasts well and leads to smearing and migration smiles due to the limitations of ray theory.

Use of Kirchhoff migration will also probably affect the velocity analysis.

Redoing the PSDM with with reverse-time migration or even a one way wave based shot migration would probably give a better result. At least a test of a part of the data volume could be performed to investigate the sensitivity due to choice of migration algorithm.

We agree with the reviewer that the 3D Kirchhoff PSDM could produce some smearing and smiles, especially in the case of high velocity contrasts, such as at the seafloor and in the melt sill region (magma domain). However, the main focus of this paper is the lava flow layering, which seem to be associated with low velocity contrasts, evidenced by the weak reflection amplitudes. Moreover, even though these layers are dipping, they are nearly sub-

horizontal locally, hence the result should not be affected by the type of migration algorithm used. In order to see the effect of aperture in the crossline direction, we performed PSDM using 1 km, 2 km and 4 km aperture, and results are shown in Supplementary Figure 16. We find that when the aperture is small (e.g., 1 km), the smiles are less pronounced but there is smearing (Supplementary Figure 16a). On the other hand, smiles are more pronounced from the melt sills when the aperture is large (Supplementary Figure 16c), but the images of the melt sills and LAB are more focused, and are less smeared. So there seems to be a trade-off between the smearing and smiles. As our focus here is layering above the magma domain, we preferred to interpret the 1-km aperture image here. We would very much like to perform 3D RTM, but we would like to get much better velocity model from 3D full waveform inversion before performing 3D RTM; a proposal is under review for funding.

We noticed that the smearing and smiles are more pronounced in the cross-line direction, possibly due to 37.5 m line spacing, therefore, we performed 2D RTM along the same profile as used for the FWI (Supplementary Figure 20) and compared with our 3D Kirchhoff results (Xline 2448), shown in Supplementary Fig. 19. The 2D Kirchhoff image (Supplementary Figure 19a) shows weak layered reflections above the magma domain but these features are more pronounced in the 2D RTM image, but comparable to the 3D Kirchhoff image (Supplementary Figure 16a).

2D FWI

The 2D Full Waveform Inversion workflow is adequate and the results seem to be sensible. Also the residuals are substantially reduced from first to last iteration confirming convergence.

However, although residuals are small, resolution might still suffer.

Some kind of resolution analysis would be good for verifying the quality.

Thanks to the reviewer for pointing out about the resolution test. Yes, we always do resolution tests, but felt that it was not so important in this paper. In this revision phase, we have performed checkerboard tests using checker sizes, 1.5 km x 1 km, 1 km x 1 km, and 0.5 x 0.5 km shown in Supplementary Fig. 24. These results suggest that we can resolve structures of the size 0.5 km x 0.5 km in the upper crust, and have discussed these details in the Methods section.

REVIEWERS' COMMENTS

Reviewer #1 (Remarks to the Author):

Review of revised “Oceanic upper crustal accretion by melt sill and lava flow interaction at Axial volcano”

The manuscript presents an interesting view of the crustal architecture at Axial, a type section of hotspot-influenced spreading ridge, and provocative interpretation of the magma delivery system that differs from the standard layered model. The key new contribution is the imaging of layering within the volcanics. This is new and is significant to the volcanology community. The manuscript has been revised addressing reviewers' comments. I find the sections on Magma Domain, Subsidence, and Hydrated-Dehydrated Boundary are very well supported. The section on Assimilation is a bit speculative and it is didn't see the point of a long-ish discussion of this topic given the previous comments by reviewers 1 and 2. This is the one area where I thought their conclusions were not strongly supported by the data. The new explanation of the trade-offs with the Kirchhoff migration seem to justify it well, and although reviewer 3 is technically correct that RTM could produce cleaner imaging, I find the images presented are clear enough to interpret easily. I find the methodology to be sound and reported in sufficient detail.

Lines 51-53: I agree that there is an unnecessary amount of jargon/detail here about “stacking a retrograde refraction/reflection branch” followed by Figure 1a which does not enlighten the reader about branches but does show layer 2A. Most people aside from seismologists would probably be happier with the layer 2A event being a reflection produced in a high velocity gradient zone.

We have taken the two reviewers comments into account and have removed the jargon, and wrote as seismic Layer 2A/2B boundary results from the stack of wide-angle data.

Line 105: I was expecting a figure that showed the LAB geometry, and I feel like a better choice could be made than Figure 2.

We have now referred to Figs. 3 and 4 as well.

Line 159: I'm not sure what “baseline” lava flow layers are, or how they differ from other lava flow layers.

'Baseline' has been replaced by 'typical'.

Line 184: It's unclear how “lava flow units” are defined, but I can see that the volcanic pile does thicken overall as toward the rift zones and magma domain.

We have replaced 'lava flow units' with 'volcanic piles', which is a better representation.

Reviewer #2 (Remarks to the Author):

Review of revised manuscript by William Chadwick

General comments on the manuscript:

This is my second review of this manuscript, which has been substantially revised. In general, the revisions have addressed most of my previous comments and concerns. However, the paper would benefit from another round of minor revision before publication. I have made suggested edits and some additional comments in an annotated manuscript Word file (with the figures removed to reduce the file size). A few of the more significant comments are included below.

Thanks to the reviewer for providing detailed comments and edits in the manuscript, which we have tried to address as best we can.

Specific comments (keyed to lines in the annotated manuscript):

Line 100: Descriptions of what is shown in the Movies needs to be added to the Supplementary Materials file. There are currently no descriptions or explanations at all. For example, what is the colored surface that is added below the seafloor in Movies #3, #4, and #5?

Nature Communications required the Movie Legends be included in the Cover Letter, which we did. In the revised Description, we have included details about the colour surface. Now we have included separate files for each Movie Legend.

Line 202: This sort of geometry is observed on land in Iceland (lava flows exposed on the east and west coasts dip toward the spreading plate boundary). You should highlight this similarity more since it seems very relevant. For example, this paper discusses the same mechanisms of spreading and subsidence leading to “inward dipping” lava flow stratigraphy:

Daignieres, M., V. Courtillot, R. Bayer, and P. Tapponnier (1975), A model for the evolution of the axial zone of mid-ocean ridges as suggested by Icelandic tectonics, Earth Planet. Sci. Let., 26, 222–232.

We have rewritten this paragraph to emphasise the link between lava flow layering observed at Axial and with those observed in Iceland, and have included the above reference.

The rest of this sentence I don't understand. What difference does it make whether a MOR is subaerial or submarine in regards to this volcano-tectonic process? And the seaward dipping reflections (SDRs) on continental margins seem like a completely different thing and NOT relevant to this discussion, so I would omit that.

The sub-aerial and sub-marine magmatic processes tend to be different because of rapid cooling and entrapment of water instead of air and efficient hydrothermal circulation in marine environment. However, we have removed this sentence. SDRs are formed by similar process and we have kept in the text.

The similarities between the cross-axis structure in Iceland and the inward-dipping

lava flows you see at Axial deserve a more coherent discussion here.

We have done that; thanks

Reviewer #3 (Remarks to the Author):

My comments on the full waveform inversion has been taken into account, a resolution test has been performed and adequately discussed. The issue with migration artifacts still remains, but the authors discussion has clarified the issue, and it is not likely that main results of the manuscript will be significantly changed with a more adequate migration algorithm.

We agree with the reviewer. Hopefully, we shall have RTM image in the future.